# Strong polar vortex favoured intense Northern European storminess in February 2022

Ryan S. Williams [1,2] ✉, Amanda C. Maycock [1], Vincent Charnay[1,3], Jeff Knight [4] & Inna Polichtchouk[5]

February 2022 was an unusually stormy month over Northern Europe, including three extratropical cyclones impacting the United Kingdom and Ireland within a single week. The month also experienced an exceptionally strong stratospheric polar vortex; however, the role of this in preconditioning the risk of extratropical cyclone hazards has not been explored. Here we use constrained subseasonal forecasts to isolate the effect of the strong stratospheric polar vortex on the North Atlantic storm track in February 2022. We estimate the strong polar vortex led to a 1.5-3-fold increase in the likelihood of a cyclone with comparable intensity to the most intense storm that impacted the United Kingdom. We also show an increased likelihood of 3 or more storms reaching the United Kingdom in a single week by ~80% compared to if the polar vortex had been of average intensity. Using a storm severity index, we estimate a 3-4-fold increase in wind gust hazards over Scandinavia and Scotland and increases in monthly precipitation over Scotland, northern England and Ireland, and Scandinavia. The results show that the strengthened stratospheric polar vortex enhanced the risk of extreme North Atlantic extratropical cyclones, serial cyclone clustering, and their associated impacts over northern Europe in February 2022.

February 2022 was notable for its intense cyclonic activity across the northern North Atlantic, affecting the United Kingdom (UK), Ireland, Scandinavia and Germany (Fig. 1a). For the first time since 2015, when the UK Met Office began naming impactful extratropical cyclones, the UK experienced three named storms in a week (Dudley, Eunice and Franklin), constituting a serial cyclone clustering event[1,2]. A total of 7 storms were tracked near the UK during the month, the 4th highest number during February since 1979 (Fig. 1b). Storm Franklin attained the strongest intensity, with a minimum central mean sea level pressure (MSLP) near the UK of 954 hPa (see "Methods"). Associated with these storms, north-west Europe experienced extreme near-surface wind gusts and higher precipitation totals than average for February. Parts of the UK, Ireland, the Netherlands, Germany and Poland experienced monthly maximum 10 metre (m) wind gusts >10 m s$^{-1}$ higher than average for February (Fig. 1c). Across much of the UK, the monthly February rainfall was more than 50% above average and was 100% higher than normal in countries bordering the Baltic Sea (Fig. 1d). Four deaths were reported across the UK and Ireland and >1 million homes were affected by a power outage that lasted several days[3]. The estimated insured losses in the UK and the rest of Europe due to

the windstorms was €3.8bn[4]. Storm Eunice, which impacted the UK on 18th February 2022, was described as a once in a decade storm and one of the most severe since 2014[5,6]. Eunice underwent explosive cyclogenesis, leading to the formation of a sting-jet feature[7,8] and an England record wind gust of ~54.5 m s$^{-1}$ (122mph) at the Needles on the Isle of Wight. The rapid development of Eunice has been attributed to its interaction with the strong jet stream, which was in excess of 90 m s$^{-1}$ (200 mph)[3,9].

The wintertime Arctic stratospheric polar vortex (SPV) is an isolated, cold polar air mass surrounded by a band of strong westerly winds between 10 and 50 km altitude[10]. Alongside the anomalous surface North Atlantic conditions, February 2022 exhibited an unusually strong SPV, with extratropical lower stratospheric (50–70°N, 100 hPa) zonal mean zonal wind anomalies of 7–10 m s$^{-1}$ above the long-term climatology (Fig. 2a, b), The monthly mean SPV was the second strongest since 1979, well above the 90th percentile (Fig. 2c).

The strong SPV in this period has been linked to suppressed vertical wave propagation into the stratosphere due to the La Niña state, a negative phase of the Pacific Decadal Oscillation and anomalously warm sea surface temperatures in the North Pacific[11]. SPV anomalies are known to impact

[1]School of Earth and Environment, University of Leeds, Leeds, UK. [2]British Antarctic Survey, Cambridge, UK. [3]Antarctic Research Centre, Victoria University of Wellington, Wellington, New Zealand. [4]Met Office Hadley Centre, Exeter, UK. [5]European Centre for Medium-Range Weather Forecasts (ECMWF), Reading, UK. ✉e-mail: rywill@bas.ac.uk

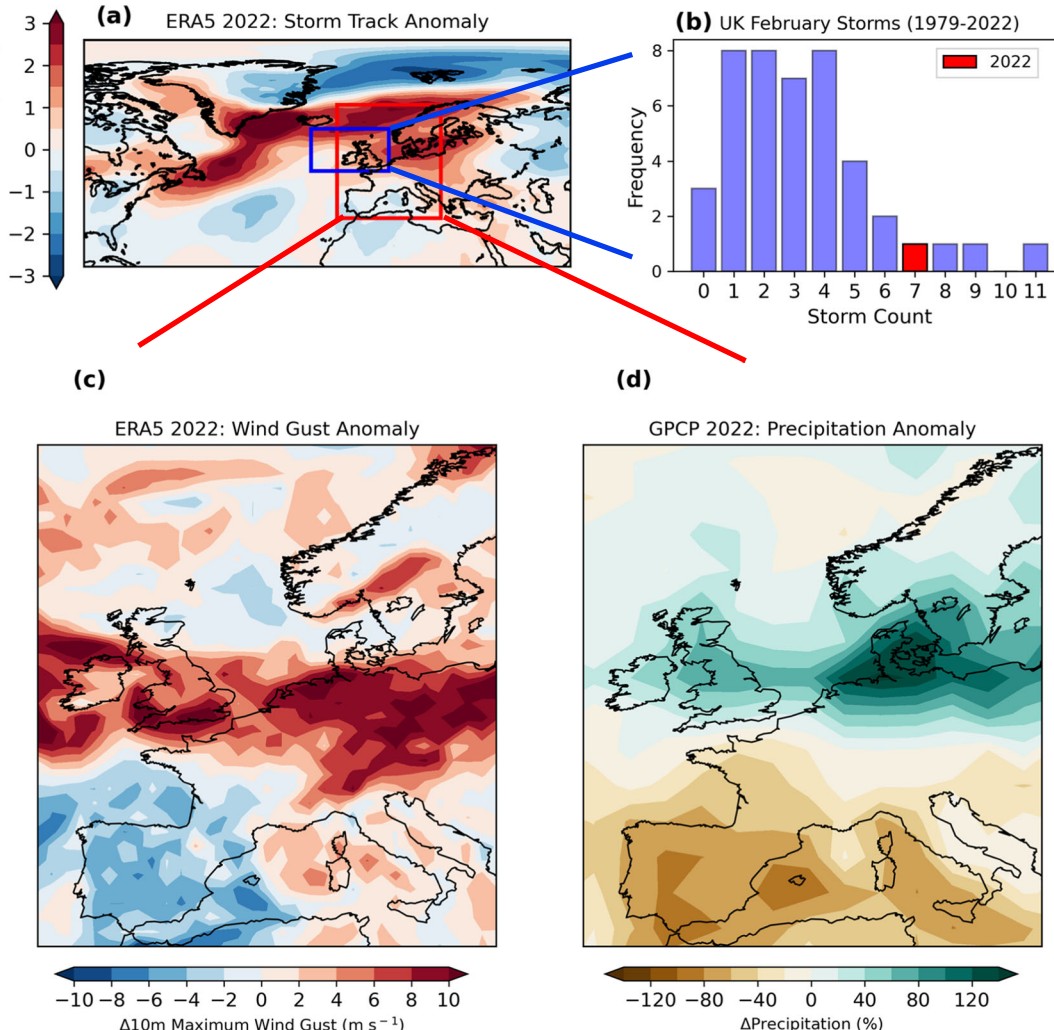

**Fig. 1 | Observed cyclonic conditions and associated impacts over the North Atlantic and western Europe during February 2022. a** Anomaly in 2-6-day bandpass filtered February 2022 MSLP variance to highlight the location and intensity of the preferential storm track across the North Atlantic and Europe during this month; (**b**) Histogram of February monthly total storm counts (1979–2022) passing over the UK region according to ERA5 (2022 in red); (**c**) Map of monthly maximum 10 m wind gust (m s$^{-1}$) anomalies for central-western Europe and (**d**) Same as (**c**) but for the monthly total precipitation anomaly (%). Anomalies are all relative to the 1979–2021 baseline. The blue box in (**a**) denotes the UK domain used in the cyclone analysis shown in panel (**b**). The red box in (**a**) denotes the region shown in panels (**c**) and (**d**).

surface weather and climate through stratosphere-troposphere coupling, particularly over the North Atlantic sector in late winter[12]. On average, a strong SPV coincides with a positive phase of the North Atlantic Oscillation (NAO)[13,14], a poleward shifted storm track[15,16], and a decrease in average North Atlantic cyclone minimum MSLP[15]. Given that strong SPV conditions often persist for several weeks or longer, this could offer a source of subseasonal predictability for the position of the storm track and main regions of cyclogenesis[15,17]. The influence of a strong SPV on the jet stream could also affect serial cyclone clustering, which has been linked to a persistent, zonally orientated and intensified jet over the eastern North Atlantic[2]. A strong SPV was suggested as contributing to the succession of cyclones that impacted north-west Europe during February 2020[18,19], although at the time of writing a detailed attribution of the stratospheric influence on this event has not been performed. Therefore, what influence the SPV has on extratropical cyclone clustering and associated weather hazards remains an important unanswered question[20].

This study investigates the role of the strong SPV on the anomalous North Atlantic storm track in February 2022 using operational multi-model ensemble seasonal forecasts (C3S), separated into ensemble members that simulate a strong and average strength SPV (see "Methods"; Fig. 2a). Additional sets of subseasonal reforecasts using the Met Office GloSea6 and

ECMWF IFS systems were performed to isolate the stratospheric influence by using an atmospheric nudging technique that relaxes the stratospheric state to either reanalysis data for February 2022 or a long-term climatology[21] (see "Methods"). These simulations reproduce the target extratropical lower stratospheric winds closely, and show broadly similar signals, so for brevity the two systems are aggregated (SNAPSI; Fig. 2b), including for some later results. We compare the characteristics of North Atlantic extratropical cyclones in the pairs of forecasts to determine the influence of the strong SPV on the storm track, including cyclone maximum intensity, serial cyclone clustering and surface hazards.

The strong SPV in February 2022 induces a poleward shifted, intensified North Atlantic jet stream (Fig. 3a–c; Fig. S1), an MSLP anomaly pattern that resembles a positive NAO index anomaly (Fig. 3d–f; Fig. S2), and an enhanced and poleward shifted storm track from Newfoundland to the Norwegian Sea (Fig. 3g–i; see also Fig. S3). The modelled NAO index anomaly associated with the strong SPV signal is ~+1.5–1.7 (Fig. 3e, f) close to that in the reanalysis (+2.05; Fig. 3d; see "Methods"), which was the 6th most positive value for February since 1979. In summary, the C3S and SNAPSI experiments reproduce the canonical North Atlantic monthly average circulation response to a strong SPV[22], which broadly resembles the signal of a weak SPV but with the opposite sign[23].

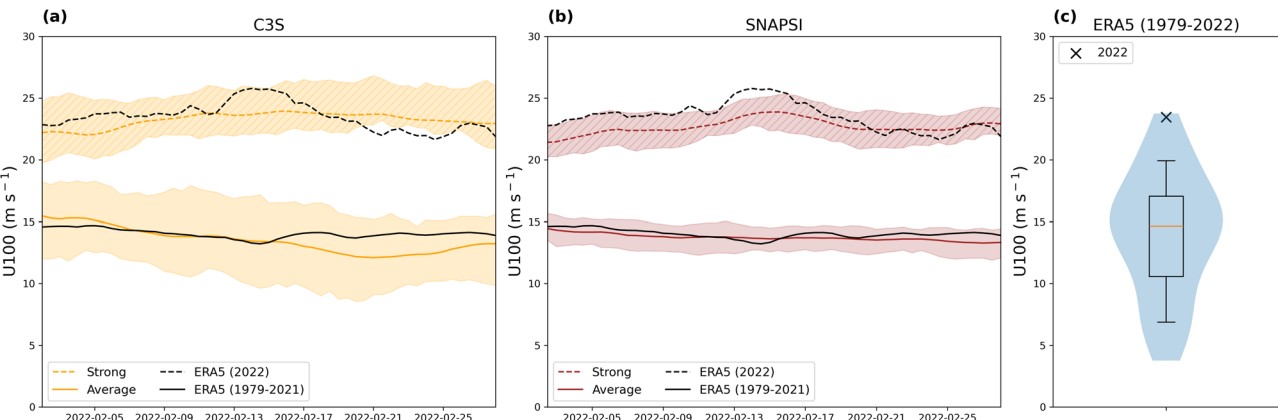

**Fig. 2 | Anomalously strong extratropical lower stratospheric winds in February 2022.** 12-hourly evolution in 100 hPa zonal mean zonal wind (U100) averaged from 50 to 70°N. **a** C3S Strong SPV (dashed orange line) and Average SPV (solid orange line), with the 25th–75th percentile range of contributing members in shading (hatched for the Strong members). See "Methods" for description of member selection approach. **b** As in (**a**) but for the Strong SPV (brown dashed line and hatched shading) and Average SPV (brown solid line and shading) SNAPSI (GloSea6 and IFS models combined) experiments, respectively (see "Methods"). **c** Violin plot of the ERA5 February mean wind strength (1979–2021), with February 2022 indicated (black cross). Orange line shows the median, the box shows the interquartile range, whiskers show 10th–90th percentile range and shading the maximum/minimum limits.

**Fig. 3 | Tropospheric signature of the strong SPV influence over the North Atlantic and Europe (20-90°N; 90°W-90°E).** February 2022 anomalies in (**a**–**c**) monthly mean upper tropospheric (300 hPa) wind speed (shading) versus climatological values (solid contours) for ERA5, C3S and SNAPSI. **d**–**f** Anomalies (shading) in monthly mean MSLP relative to climatology (solid contours) for the three datasets. The monthly mean station-based NAO index anomaly (see "Methods" for calculation) is shown for each dataset. **g**–**i** The monthly mean anomaly in 2–6-day bandpass filtered MSLP variance (see "Methods") for each dataset. Anomalies are calculated with respect to 1979–2021 for ERA5 and as the ensemble-mean difference between the Strong and Average SPV state for C3S and SNAPSI. Stippling or hatching denotes statistical significance at the 95% confidence level using a paired Student's t test.

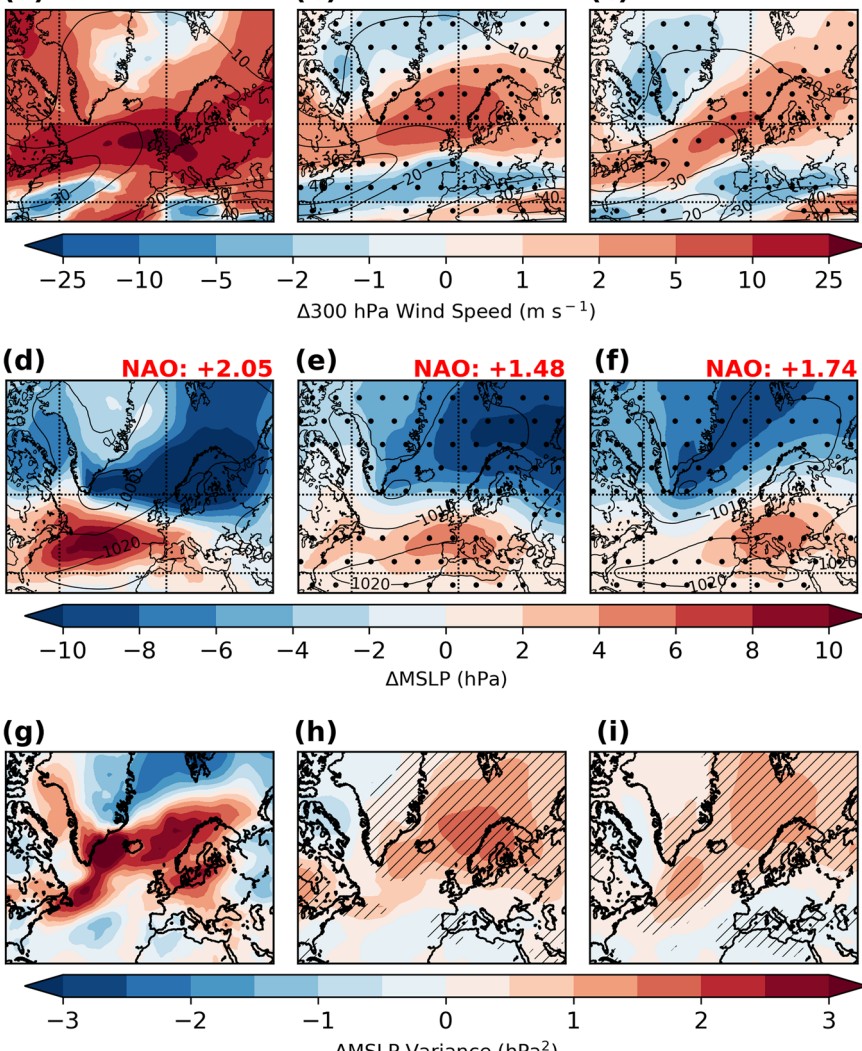

## Results

### SPV influence on storminess

The C3S and SNAPSI experiments show enhanced cyclonic activity in the northern North Atlantic, Northern Europe and Scandinavia (Fig. 3g–i; Fig. S3) and an increased risk of high intensity cyclones reaching the UK in February 2022 due to the strong SPV (Fig. 4). The median minimum cyclone central MSLP over the UK region decreases by an average of −5.1 (−1.8 to −8.3) hPa (Table 1; range indicates the minimum and maximum shift across the model datasets). This shift in median intensity corresponds to around the 35th (28th–44th) percentile of the average SPV reference distribution. We note the majority (>50%) of the decrease in cyclone central MSLP over the UK remains after subtracting the background MSLP from the reference simulation along each track (Fig. S4 and Table S1). This demonstrates that the strong SPV favoured stronger cyclogenesis and is not merely due to a northward shift of the storm track to a region of lower ambient pressure[15].

Notably for C3S and GloSea6, the shift to deeper cyclones associated with the strong SPV is larger for lower percentiles of the distribution, representing an elevated relative risk (RR; see "Methods") for increasingly intense cyclones (Fig. 4d). Under strong SPV conditions, the RR of a cyclone corresponding to the 25th percentile of cyclone central MSLP increases by an average of 25% (10–46%) and by 68% (31–92%) for the 5th percentile, relative to average SPV conditions. Importantly, this means that although the deepest observed cyclone reaching the UK had a central MSLP of 954 hPa (Storm Franklin), corresponding to the ~9th (3rd–13th) percentile, there was a substantially elevated risk of an even more intense cyclone impacting the UK due to the strong SPV. We note that an enhanced RR is seen for the IFS model down to a minimum central MSLP of ~945 hPa, but at lower pressures the RR begins to return towards 1. This may be because, climatologically, IFS produces deeper cyclones than found in the reanalysis[24] (Fig. S5), which might limit the influence the strong SPV has on the most intense cyclones in this model. Although the differences in minimum central MSLP over the UK are smaller in C3S than in the SNAPSI models (Fig. 4a–c; Table 1), they are comparable between datasets over the wider North Atlantic sector (Fig. S6, Table S2). This suggests there is a similar relationship between the strong SPV and cyclone intensity in the C3S models, but more cyclones track to the north of the UK region (Fig. S3).

We next consider whether the strong SPV impacted the likelihood of serial cyclone clustering, given the unusual, quick succession of observed storms Dudley, Eunice and Franklin. We quantify the rolling weekly count of cyclones which intersect the region surrounding the UK and reach an intensity of <970 hPa during their lifetime (see "Methods"). A lower central MSLP threshold is used compared with the cyclone statistics in Fig. 4 because serial clustering is most commonly associated with intense cyclones[25–27]. The analysis shows the strong SPV increased the likelihood of 3 or more cyclones impacting the UK region within one week by around 80% compared to if the SPV had been average (Fig. 5; increased likelihoods of 56% for C3S, 113% for GloSea6 and 92% for IFS). Both the higher cyclone intensity and the increased likelihood of cyclone clustering under strong SPV conditions are likely to be linked to the strengthened upper-level jet stream (Fig. 3a–c; Fig. S1), which has been shown to promote cyclogenesis[20,28].

### Influence of the strong SPV on cyclone hazards

The risk of land-based hazards from intense wind gusts and heavy precipitation was elevated across Northern Europe in February 2022 because of the strong SPV (Fig. 6). There is a north-south (windier-wetter and calmer-drier) divide in the signal from the strong SPV across Europe, broadly extending from the southern UK towards northern Italy (Fig. S7). For Scotland, Northern England/Ireland, Scandinavia, and Northern Germany/Netherlands (see Fig. S8 for definitions of regions), the ensemble-mean monthly maximum February 10 m wind gust increases with a strong SPV compared to if the SPV was average (Fig. 6; also Fig. S9), in qualitative agreement with the reanalysis (Fig. 1c; Fig. S9). The increase in wind gusts is between 10 and 13% for Scotland, 4–9% for Northern England and Ireland, 12–17% in Scandinavia, 2–6% for Northern Germany and the Netherlands, with a 6–8% reduction in Iberia. These changes are likely to be important for windstorm damage because of the cubic dependency on wind speed. The cumulative monthly and regional storm severity index (SSI; see "Methods") is between 2 and 4 times higher in Scotland, Northern England and Ireland, and Scandinavia under the strong SPV conditions (Table S3). The increase in the SSI is statistically significant for GloSea6 and IFS in all regions, except Iberia (here there is no change or a reduction in SSI). Whilst the signal is not highly statistically significant for the C3S models, the degree of overlap between the confidence intervals is negligible for Scotland and Scandinavia, so the signal in these regions is unlikely to be due to chance.

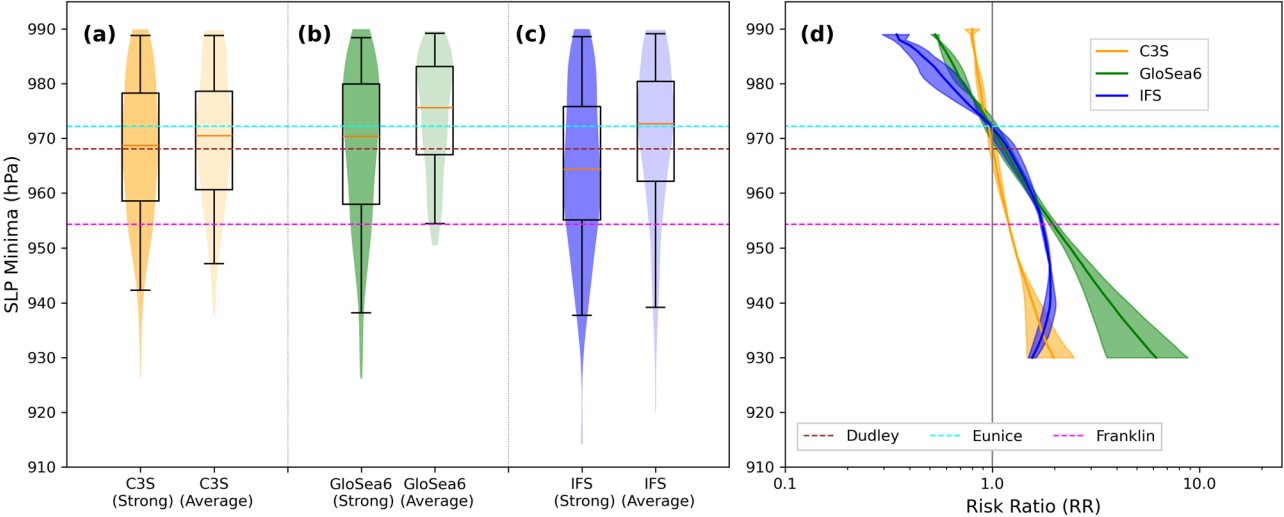

**Fig. 4 | Higher maximum cyclone intensities over the UK region during February 2022 due to the strong SPV conditions.** Maximum cyclone intensities (minimum MSLP) over the UK region (blue box in Fig. 1a) for (**a**) C3S, (**b**) GloSea6 and (**c**) IFS models under Strong and Average SPV conditions. Whiskers show 2.5th to 97.5th percentile range. **d** Relative Risk ratio (RR) curves for storms of a given intensity (MSLP minimum) between Strong and Average SPV conditions, as estimated according to a generalised extreme value (GEV) fitted distribution. Shading denotes the 95th percentile confidence interval range of each RR curve, bootstrapped over 10^4 iterations (see "Methods"), and the median is represented by the solid line. Note that this is not extended for maximum storm intensities <930 hPa due to increasingly small sample size and greater uncertainty due to data extrapolation. Horizontal dashed lines correspond to the minimum MSLP values of the three observed named storms in February 2022. Equivalent distributions in (**a**–**c**) for the wider North Atlantic sector are shown in Fig. S6.

**Table 1 | Strong SPV impact on UK storm statistics**

| Dataset | | Frequency (month⁻¹) | Cyclone Minimum MSLP [hPa] | | | | |
|---|---|---|---|---|---|---|---|
| | | | 5th | 25th | Median | 75th | 95th |
| ERA5 | Climatology | 3.2 | 947.3 | 958.3 | 971.4 | 978.8 | 986.2 |
| C3S | Strong | 2.8 | 946.4 | 958.6 | 968.7 | 978.3 | 987.9 |
| | Average | 3.1 | 949.5 | 960.7 | 970.5 | 978.6 | 988.3 |
| | Strong—Average | -0.4 | −3.1 | −2.1 | −1.8 | −0.4 | −0.4 |
| GloSea6 | Strong | 2.9 | 943.8 | 958.0 | 970.4 | 980.0 | 988.0 |
| | Average | 2.9 | 955.8 | 967.0 | 975.6 | 983.1 | 988.4 |
| | Strong—Average | 0.0 | −12.0 | −9.0 | −5.3 | −3.2 | −0.4 |
| IFS | Strong | 3.5 | 939.1 | 955.1 | 964.4 | 975.8 | 986.5 |
| | Average | 2.8 | 943.6 | 962.2 | 972.7 | 980.4 | 988.5 |
| | Strong—Average | +0.7 | −4.5 | −7.1 | −8.3 | −4.6 | −2.0 |

Cyclone frequency and cyclone minimum MSLP statistics for the UK domain during February for ERA5 (1979–2021), C3S, GloSea6 and IFS Strong and Average SPV states.

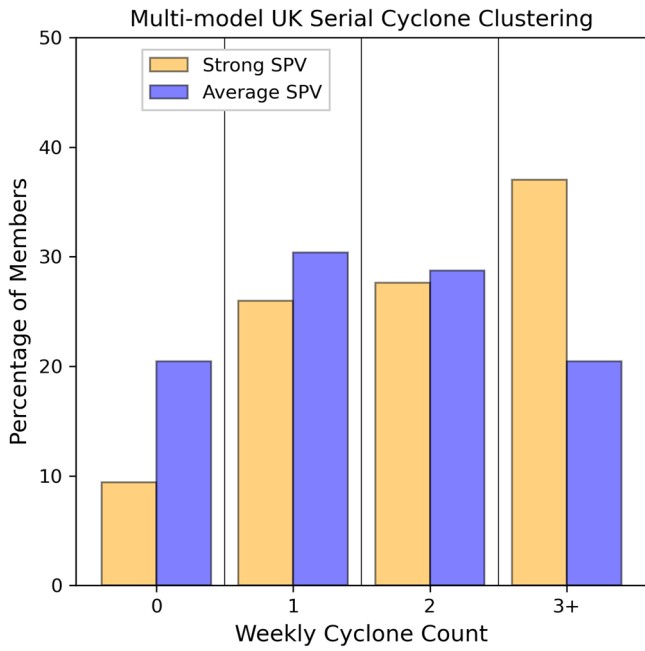

**Fig. 5 | Enhanced occurrence of UK serial cyclone clustering events in February 2022 under strong SPV conditions.** Histogram of maximum 7-day count of intense cyclones impacting the UK (see "Methods") expressed as a percentage of members. The C3S, GloSea6 and IFS members are combined for each Strong and Average SPV sample ($n = 181$).

These results show the strong SPV state significantly increased the risk of intense wind gusts across much of Northern Europe. In February 2022, the largest monthly maximum wind gust anomalies occurred within a swathe extending across Ireland, northern England, the Netherlands, and northern Germany (Fig. 1c), with widespread anomalies >8 m s⁻¹ and a relative increase of >40 % over Northern Germany and the Netherlands (Fig. S9). This represents a shift in peak wind gust intensities from gale-force (Beaufort Force 9), equating to the likelihood of slight structural damage, to violent storm (Beaufort Force 11), translating to widespread damage (Fig. S9). Our model results show a greater increase in peak wind gusts over Scotland and Scandinavia compared to Ireland, Northern England, the Netherlands and Northern Germany. This suggests the increased risk of damaging wind gusts due to the strong SPV in February 2022 was largest in those regions, despite the fact this did not coincide with the strongest observed gusts (Fig. 1c).

Monthly total precipitation also increases in several regions because of the strong SPV, including wetting over Scotland (up to ~30%), northern England and Ireland (~0–20%) and Scandinavia (~10–40%), consistent with above average precipitation in those regions seen in observations (Figs. 6, S7e–h, S10). We attribute these increases to the higher cyclone intensity and frequency in those regions resulting from the strong SPV, given the dominant role of extratropical cyclones for winter northern European precipitation[29]. We note that flooding affected each of these regions in February 2022, in part due to the above average precipitation and the antecedent hydrological conditions[9].

## Discussion and conclusions

Our results show that the strong SPV conditions in February 2022 contributed to the intense northern North Atlantic storminess, increasing the likelihood of serial extratropical cyclone clustering and associated weather hazards across Northern Europe. The model forecasts also demonstrate a stratospheric contribution to potential predictability during this period, as noted before for the North Atlantic region[30–32]. The SNAPSI reforecasts show a 16–17% reduction in root mean square error for monthly North Atlantic MSLP and a decrease in ensemble spread by ~10–12% for the experiments constrained to follow the observed strong SPV state (Fig. S11). While it was not the focus of this study, it is notable that the C3S models initialised on 1 January 2022 were confident the February SPV would be stronger than average, since the average SPV forecasts shown in Fig. 2a correspond to the weakest 20% of members. The signal for a strong SPV was evident from forecasts initialised as early as November 2021 using GloSea6[33]. This suggests that strong SPV states could offer a 'window of opportunity' for enhanced European predictability, particularly over Central and Southern Europe[34]. We note that the Madden-Julian Oscillation (MJO), another known driver of North Atlantic subseasonal climate variability[35], was relatively inactive during January and early February 2022[11], meaning this would not have had a strong influence during this period.

While our results are specific to February 2022, it is plausible that other years with strong SPV conditions could exhibit a similar influence on the North Atlantic storm track. Therefore, it would be insightful to apply the attribution approaches used here to study other periods with strong SPV conditions, to complement composite approaches[15]. For example, at the time of writing, February 2020 had the strongest monthly mean SPV since 1979[36] and also exhibited intense North Atlantic cyclogenesis[37]. However, the influence of the SPV on the storm track in February 2022 may have been shaped by other factors related to the tropospheric state at the time, which are captured in the initialised model experiments. It is intriguing that both these observed anomalies occurred in February and future work might examine whether a strong SPV in early winter imparts a similar influence on

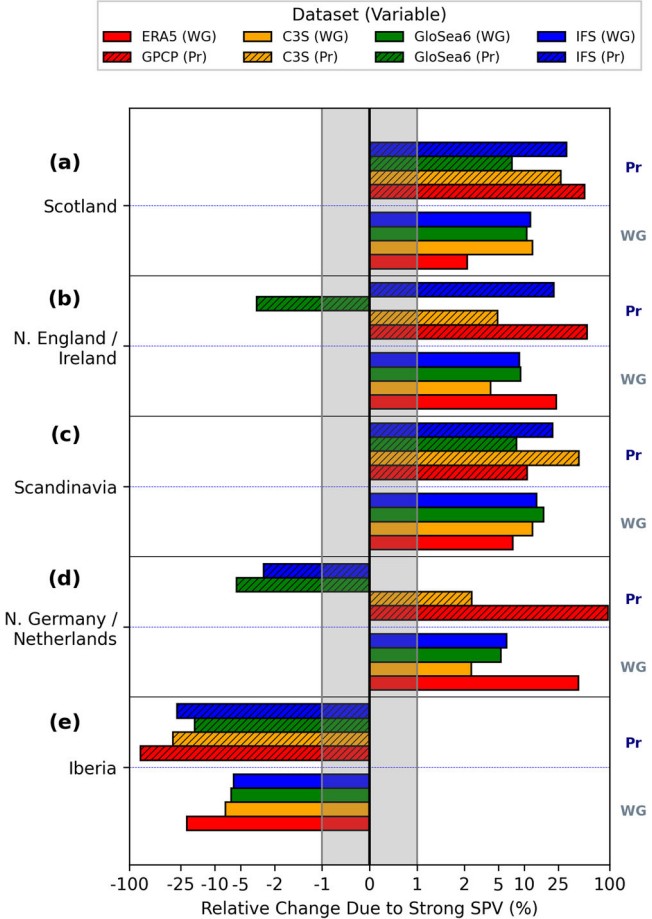

**Fig. 6 | Effect of the strong SPV in February 2022 on 10 m extreme wind gusts and monthly precipitation accumulation for different European regions.** Anomalies (%) in February monthly maximum 10 m wind gust (WG) and total monthly precipitation (Pr), computed over all land grid cells in each region for (**a**) Scotland, (**b**) Northern England and Ireland, (**c**) Scandinavia, (**d**) Northern Germany and the Netherlands and (**e**) Iberia. WG and Pr anomalies are represented by unfilled and hatched bars, respectively. C3S (yellow), GloSea6 (green) and IFS (blue) show ensemble-mean values. Observed anomalies are in red. Note that the x-axis is log-scaled except between −1 and 1% which is linearly scaled (grey shaded region).

the storm track. Future work should also investigate the influence of more moderate SPV anomalies.

Several studies have identified an increase in cyclone hazards in Europe under future climate change[38–40]. This implies greater exposure to severe storm impacts in the coming decades. Understanding subseasonal drivers of storm track variability that are potentially predictable, including the SPV, is therefore key to provide early warnings of severe weather events in Europe.

## Methods

### Reanalysis data

To estimate the observed atmospheric state during February 2022 and the preceding 43-year climatological period (1979–2021), we extract 6-hourly MSLP, 12-hourly zonal wind (u) at 100 hPa, 6-hourly zonal and meridional (u and v) wind at 300 hPa post-processed to monthly means, and the 6-hourly 10 m wind gust since previous post-processing from the fifth generation of the European Centre for Medium-Range Weather Forecasts (ECMWF) reanalysis (ERA5[41]). The monthly mean precipitation anomaly was derived from the Global Precipitation Climatology Project v2.3 (GPCP[42]), a gridded monthly precipitation dataset available at 2.5° × 2.5° resolution, covering the period 1979–2022. ERA5 data (retrieved on the native grid; 0.25° horizontal resolution) was bilinearly interpolated to the same horizontal resolution of the Copernicus Climate Change (C3S) seasonal forecast datasets (1° × 1°). In our analysis, we convert the average monthly precipitation rate (mm day$^{-1}$) to the monthly total accumulated precipitation (m).

### Seasonal forecast datasets

We use output from 7 seasonal forecast systems from the Copernicus Climate Data Store (CDS), available at: https://cds.climate.copernicus.eu/datasets/seasonal-original-single-levels?tab=overview[43] and https://cds.climate.copernicus.eu/datasets/seasonal-original-pressure-levels?tab=overview[44]. The systems are shown in Table 2. We use the operational forecasts with effective initialisation for 1 January 2022 and analyse them at a 1-to 2-month lead time. The ensemble creation methods for each system are described at https://confluence.ecmwf.int/display/CKB/Description+of+the+C3S+seasonal+multi-system. Similarly, as for the reanalysis datasets, we retrieved 6-hourly MSLP, 12-hourly 100 hPa u, and daily maximum 10 m wind gust since previous post-processing. Additionally, we acquired monthly 300 hPa u and v and daily total precipitation (mm). The 10 m wind gust since previous post-processing is the maximum 3-s wind gust simulated within each time interval (i.e., 24-h window). All variables are output

## Table 2 | C3S model description

| Centre | Ensemble size | Atmospheric resolution | Oceanic resolution |
|---|---|---|---|
| ECMWF* | 51 (25) | T$_{CO}$319/L91<br>Dynamics: T$_{CO}$319 cubic octahedral grid<br>Physics: O320 Gaussian grid (36 km)<br>91 levels in vertical, to 0.01hPa (80 km) | 0.25° ORCA grid<br>75 levels in vertical |
| UKMO* | 60 (21) | N216/L85<br>0.83° x 0.56° (~60 km in mid-latitudes)<br>85 levels in vertical, to 85 km | 0.25° ORCA grid<br>75 levels in vertical |
| MétéoFrance | 51 | TL359/L137 (0.5°)<br>137 levels in vertical, to 0.01hPa | 0.25° ORCA grid<br>75 levels in vertical |
| DWD | 50 | T127 (~100 km)<br>95 levels in vertical, to 0.01hPa | 0.4° TP04 grid<br>40 levels in vertical |
| CMCC | 50 | approx 0.5° lat-lon<br>46 levels in vertical, to 0.2hPa | 0.25° ORCA grid<br>50 levels in vertical |
| JMA | 91 | TL319 (approx. 55 km)<br>100 levels in vertical, to 0.01hPa | 0.25° tripolar grid<br>60 levels in vertical |
| ECCC (GEN5-NEMO) | 10 | ~1.1° lat-lon (~110 km)<br>85 levels in vertical, to 0.1hPa | 1/3° (equator) to 1°<br>50 levels in vertical |

The C3S seasonal forecast models used in the study. Hindcasts were also retrieved for the two asterisked centres to assess bias in cyclone distribution intensities relative to ERA5 over a common baseline period (1994–2016), shown in Fig. S5 and Table S4. The hindcast ensemble size is shown in parentheses in the next column (note for UKMO, the set of 7-member forecasts are pooled together for 25 December, 1 January and 9 January initialisation dates).

on a regular $1° \times 1°$ horizontal grid, except for the JMA model which is available on a $2.5° \times 2.5°$ grid. To avoid degrading spatial information in the C3S models, we elected to perform the analyses on a $1° \times 1°$ grid, with the exception of JMA. For precipitation, however, we conservatively remapped the model fields to match the equivalent resolution of the GPCP dataset ($2.5° \times 2.5°$ grid).

Each C3S forecast is subset individually to select the >80th percentile (Strong SPV) and <20th percentile (Average SPV) members, based on the magnitude of the February mean 100 hPa zonal mean u ($U100$) averaged over 50–70°N. The subsets of members are then pooled across models to create a multi-model ensemble of strong and average SPV states. The median strong SPV state happens to align closely with the ERA5 lower stratospheric wind in February 2022, and the median average SPV state is close to the climatological ERA5 wind (Fig. 2a). Whilst this agreement was not expected a priori, it indicates the C3S multi-model forecasts were confident that the SPV would be above average in strength at a 1- to 2-month lead time (highlighted by the fact that the weakest 20% of members correspond to an average SPV).

We additionally analyse the 23-year C3S hindcasts (1994–2016) for GloSea6 and IFS to evaluate the climatological storm characteristics against ERA5 for the same period (Fig. S5; Table S4). These models are chosen because they are used for the SNAPSI nudged subseasonal reforecasts. For GloSea6, 7 ensemble members were available for each of the following initialisation dates: 25 December, 1 January and 9 January ($n = 21$) for each year, yielding a sample size of $n = (23 \times 21) = 483$. For IFS, all 25 members were extracted on the 1 January initialisation date, giving a sample size of $n = (23 \times 25) = 575$.

### Nudged subseasonal reforecasts

Subseasonal reforecasts initialised on 1 January 2022 and integrated for 70 days are performed with the GloSea6 and IFS models. GloSea6 comprises a 59-member ensemble and IFS has 50-members (49 members for the precipitation variable due to a data issue). The reforecasts use a zonally symmetric nudging technique to constrain the stratospheric state. While nudging can be applied in 3-dimensions[21], the SPV tends to be relatively zonally symmetric when it is strong, so nudging the zonal mean state should be suitable to capture most of the signal, consistent with earlier studies[45–48]. One experiment is relaxed to the stratospheric anomalies from ERA5 (Strong SPV) and another experiment is relaxed towards the long-term ERA5 climatology (Average SPV). The nudging uses a Newtonian relaxation approach for zonal mean wind ($\bar{U}$) and temperature ($\bar{T}$) with a 6-h relaxational constant. Following the protocol described by Hitchcock et al. (2022)[21], nudging is performed uniformly over all latitudes at pressures below 50 hPa (i.e., at higher altitudes), slowly tapering off according to a cubic function below this region, such that the nudging strength is zero at pressures greater (altitudes lower) than 90 hPa. The levels impacted by the nudging were selected to avoid directly impacting the troposphere, yet to constrain the stratosphere as much as feasible. It has been verified previously that 2D nudging following this protocol is able to closely reproduce the modelled tropospheric state seen in a free-running model for specific cases[49] and does not introduce any significant artificial circulations in the free troposphere[47], which was an important precondition for undertaking this study. The tight constraint of the nudging on the stratospheric state for each model is shown in Fig. S12, including in the lower stratosphere below the nudging layer.

For all variables, the 6-hourly output (except 10 m wind gust since previous post-processing for GloSea6 which was available hourly) from the simulations were aggregated to daily resolution for direct comparison with the C3S forecasts. The precipitation flux (kg m$^{-2}$ s$^{-1}$) output for GloSea6 was, converted to total precipitation (m) to be consistent with the IFS, C3S and GPCP data. As all IFS data was output on a $1° \times 1°$ horizontal grid, consistent with the C3S forecasts, no regridding was performed. For GloSea6, the output on a $0.83° \times 0.56°$ grid was bilinearly interpolated to a $1° \times 1°$ grid. Precipitation data was conservatively remapped for both GloSea6 and IFS to $2.5° \times 2.5°$ horizontal resolution to be consistent with GPCP.

### NAO index calculation

The station-based NAO index[50] is computed as the normalised MSLP difference (Eq. (1a)), between the model grid cells closest to Lisbon, Portugal (37.71°N; 9.14°W) and Reykjavík, Iceland (64.13°N; 21.93°W). For ERA5, we first normalise the February 2022 value at each location by subtracting the mean over the preceding 43 years (1979–2021) and dividing by the standard deviation of the monthly mean MSLP over all years. For the three model datasets, we normalise the Strong SPV ensemble-mean MSLP value at each location, by subtracting the Average SPV ensemble-mean value and then dividing by the standard deviation of the monthly mean MSLP over Average SPV members (Eq. 1b, c). This is expressed mathematically as:

$$NAO_i = p(Lis)'_{Norm,i} - p(Rey)'_{Norm,i} \tag{1a}$$

$$\text{where :} \qquad p(Lis)'_{Norm,i} = \frac{p(Lis)_i - \overline{p(Lis)}}{\sigma(p(Lis))} \tag{1b}$$

$$\text{and :} \qquad p(Rey)'_{Norm,i} = \frac{p(Rey)_i - \overline{p(Rey)}}{\sigma(p(Rey))} \tag{1c}$$

where $p$ is the February monthly pressure at locations Lisbon *(Lis)* and Reykjavík *(Rey)*. Primes denote anomalies for month $i$, overbars represent the climatology (for ERA5 over the period 1979–2021 and for the models using the Average SPV members), and sigma represents the February standard deviation over all years (ERA5) or all Average SPV members (C3S and SNAPSI).

### Extratropical cyclone tracking

For North Atlantic extratropical cyclone detection and tracking, we use the open-source TempestExtremes (TE) v2.2 software. TE is a framework for multifaceted atmospheric feature detection and tracking[51–53]. The software is comprised of two programmes: DetectNodes and StitchNodes. DetectNodes identifies MSLP minima (closed contours) at each timestep. We apply filtering to ensure that all candidate points are merged within a 6° great circle distance, such that only the lowest value is retained. This is provided that candidate points are enclosed by a closed contour of at least 2 hPa difference relative to anywhere within the 6° radius from the storm centre. The output from DetectNodes is passed to StitchNodes, which connects cyclonic centres at each timestep to produce storm trajectories. Further filtering is implemented to retain only storms with a lifetime of at least 60 h (2.5 days). In assigning individual storm trajectories, additional criteria are specified such that storm centres cannot travel further than 9° great circle distance in a 6-hour period, and nodes may be absent for up to three timesteps (18 h) and still be considered part of the same system. These parameter choices are the same as those selected by Ullrich et al. (2021)[52] except that the maximum propagation distance of 9° great circle distance over a 6-hour period is 50% larger than their study. This was altered because storm Eunice travelled very quickly, and it was found that a lower threshold caused Eunice to split into two separate tracks. We additionally apply filtering of weak troughs as those in which the central MSLP remains >990 hPa throughout the system's lifetime, consistent with previous studies[15]. We note that a variety of methods exist for tracking extratropical cyclones which can produce different results[54], including in relation to cyclone clustering[55]. However, we use the same method applied to all datasets so the data are directly comparable.

As a first step, we performed cyclone tracking of in the ERA5 reanalysis between 1979 and 2022 output on a $1° \times 1°$ grid. A comparison of the minimum MSLP along cyclone tracks during February in GloSea6, IFS and ERA5 are shown in Fig. S5 for the common period 1994–2016. Over the UK domain (49.5–62.5°N; 20°W–4°E), GloSea6 shows fewer cyclones with minimum pressures below 950 hPa, while IFS shows a higher occurrence of these cyclones compared to ERA5. For the North Atlantic domain (50–70°N; 90°W–40°E), the lowest MSLP percentiles in GloSea6 and ERA5

are more comparable, but IFS still shows a higher occurrence of intense cyclones (Fig. S5b). In this analysis, we compare the cyclone characteristics within each model relative to its own reference state.

## Storm track diagnostics

Using the cyclone tracks, we compute cyclone density anomalies in 5° longitude x 5° latitude bins (Fig. S3). The maximum cyclone intensity is defined as the minimum MSLP along each track. Since a change in central MSLP could be associated with a shifted storm track rather than a change in cyclone intensity, we further compute the pressure anomalies along the tracks relative to the background mean MSLP field from the Average experiment at each gridpoint (Fig. S4).

We also compute the variance of 2-6-day bandpass filtered MSLP, which has been used as a proxy for storm track activity[56,57]. A Lanczos filter with 97 weights and a window of 24 days[58] is applied at each gridpoint to obtain a 2-day and 6-day low-pass filtered timeseries. The window length is chosen due to the 70-day period of the subseasonal reforecast experiments (1 January to 12 March). We note that window size is slightly shorter than optimal (see Fig. 5 in Duchon, 1979[58]) but produces consistent results with the cyclone tracking.

Various measures have been used to quantify serial cyclone clustering[2,59]. Here, we count the cyclone footprints that pass within a region surrounding the UK (49.5–62.5°N; 20°W–4°E; Fig. 1a) within a 7-day rolling window ($n = 22$ for February), for the subset of deep cyclones in which the maximum intensity (lowest MSLP) during the storm lifetime is <970 hPa. To determine this, we define a cyclone footprint and then check for overlap of the footprint with the UK domain. Cyclone footprints have been computed in the literature using different radii from the storm centre varying from ~5 to 20 degrees[29,60,61], with both wind and precipitation features having different structures relative to the storm centre. For simplicity, we use a 10° radius for the cyclone footprint to check for interception with the UK domain. This does not mean that all modelled cyclones would generate hazards in the region, but the primary purpose of the analysis is to assess cyclone clustering over a confined geographical region.

For each member, we chose the maximum rolling weekly cyclone count within the month and then aggregate statistics across all members from each of the three model datasets (C3S, GloSea6 and IFS), yielding a total sample of $n = 181$. Note this means the maximum weekly count does not occur at the same period of the month in every member, but this is to be expected due to random internal atmospheric variability. We select the maximum rolling weekly cyclone count within the month from each ensemble member and aggregate the values across the three datasets.

## Storm severity index (SSI)

To estimate the changing risk of damaging wind gusts under a strong SPV, we calculate the storm severity index (SSI). This is adapted from Eq. 2 in Lockwood et al. (2022)[62]:

$$SSI = \sum_{i=0}^{N} \sum_{t=1}^{9} area_i \times \left( \frac{v_{i,t}}{v95_i} - 1 \right)^3 \quad (2)$$

where $v_{i,t}$ is the maximum 10 m wind gust at location $i$ within a non-overlapping 3-day interval $t$, $v95_i$ is the 95th percentile wind gust at each location, calculated using the reference Average SPV members for all days in the month, and $area_i$ is the area of the gridcell. $N$ is the number of gridcells within each defined geographic region (see below). Whilst some studies calculate the SSI along cyclone footprints to attribute losses to specific windstorms[62], here we use the Eulerian 10 m wind gust field but calculate the maximum gust at a location over non-overlapping 3-day intervals ($n = 9$ for February), to minimise the likelihood of double counting strong gusts from a single cyclone at the same location. An SSI value is calculated for each ensemble member, with summation both temporally over all 3-day intervals ($t$) and over the area of each regional domain considered: Scotland

(55.5–59.5°N; 8–1.5°W), Northern England and Ireland (52.5–55.5°N; 10.5–0°W), Netherlands and Northern Germany (51–55°N; 3–15°E), Iberian Peninsula (36–44°N; 9.5°W–0°W) and Scandinavia (55–70°N; 4–20°E). These regions are also shown in Fig. S8.

The SSI formulation follows that used in many previous studies[62,63] but uses the 95th percentile rather than the 98th percentile as a threshold. Using the 98th percentile wind speed ($v98$) for SSI has become customary, partly because in observations it approximately coincides with a 20 m s$^{-1}$ gust in mainland Europe, which is the threshold commonly used in the insurance industry for insurance payouts[63]. Nevertheless, in the context of models and reanalyses, which often exhibit biases, the equivalence of $v98$ to the insurance payout threshold is not guaranteed.

Our choice of $v95$ was motivated by the desire to look at extreme wind speeds, but also because we are comparing two relatively small samples from the February 2022 simulations and want to ensure the threshold is not determined by one or two outlier points. We note that the choice of a lower percentile threshold, combined with the cubic dependence of SSI on wind speed, would tend to give a conservative estimate of the increased risk of cyclone damage.

## Statistical methods

In Fig. 4d, we calculate Relative Risk (RR) ratio curves of the exceedance in the likelihood of a storm of a given intensity due to the strong SPV, with respect to an average SPV. The RR curves are plotted as a function of cyclone maximum intensity (minimum MSLP) during the storm lifetime when tracking across the UK domain (49.5–62.5°N; 20°W–4°E; see blue box in Fig. 1a). For each dataset (C3S, GloSea6 and IFS), these are calculated by differencing each pair of generalised extreme value (GEV) distributions fitted to the series of all storms tracked during February 2022 (Strong versus Average SPV), across all ensemble members. The RR ratio curves are calculated according to the equation below:

$$RR(p) = \frac{1 - CDF_{GEV, Strong(p)}}{1 - CDF_{GEV, Average(p)}} \quad (3)$$

where $CDF$ is the cumulative density function as estimated using a GEV fit and $p$ is the storm central MSLP minimum value during the storm lifetime within the UK domain, where: 930 hPa $< p \leq$ 990 hPa. To quantify the effect of sampling uncertainty on the estimated RR, we again perform bootstrapping (sampling with replacement) over the series of storms tracked during the month, aggregated over all ensemble members, using $10^4$ iterations, then applying a GEV fit to each bootstrapped distribution. We present the 95% intervals of RR for a given cyclone intensity (shading in Fig. 4d) and take the median to be the central estimate of RR (solid lines in Fig. 4d).

To estimate the effect of sampling uncertainty on the SSI values in Table S3, we perform a standard sampling bootstrapping with replacement, using $10^4$ repeat samples to estimate 95% confidence intervals. For each dataset (C3S, GloSea6 and IFS), we consider there to be a statistically significant increase in the accumulated exposure (as a function of both area and exceedance above the $v_{95}$ threshold) to damaging wind gusts under a strong SPV state where the calculated confidence intervals do not overlap between ensemble pairs (Strong and Average SPV).

## Data availability

ERA5 and the C3S forecasts are available from the Copernicus Climate Data Store (https://climate.copernicus.eu/seasonal-forecasts). The GloSea6 experiments are available on request. The IFS experiments are retrievable from: https://apps.ecmwf.int/ifs-experiments/rd/i4gn/ (simulation relaxed to the observed state) and: https://apps.ecmwf.int/ifs-experiments/rd/i5a5/ (simulation relaxed towards climatology).

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

## Acknowledgements

R.S.W. and A.C.M. acknowledge funding from the NERC StratClust project (NE/X011933/1). V.C. acknowledges funding from a grant to A.C.M. from the Leverhulme Trust. J.K. was supported by the Met Office Hadley Centre Climate Programme funded by DSIT. We are grateful to Matthew Priestley for helpful suggestions on calculating the Storm Severity Index for the models.

## Author contributions

A.C.M. conceived the study and secured the funding. R.S.W. performed the analysis and produced the figures with guidance from A.C.M. and J.K. V.C. performed the initial analysis of the C3S data. J.K. ran the GloSea6 experiments and contributed to the interpretation of results. I.P. ran the IFS experiments. R.S.W. and A.C.M. wrote the manuscript with input from all authors.

## Competing interests

The authors declare no competing interests.
