## [Transparent Peer Review file · Communications Earth & Environment]

Strong polar vortex favoured intense Northern European storminess in February 2022

Corresponding Author: Dr Ryan Williams

Version 0:

Decision Letter:

Dear Dr Williams,

Your manuscript titled "Strong polar vortex favoured intense Northern European storminess in February 2022" has now been seen by 3 reviewers, and we include their comments at the end of this message. They find your work of interest, but some important points are raised. In particular, Reviewers 1 and 2 comment that some added discussion on generalizability (beyond February 2022) of results would be helpful. Reviewer 3 also request some clarification of how your results relate to existing work, as well as definition of the weak polar vortex within your simulations. We are interested in the possibility of publishing your study in Communications Earth & Environment, but would like to consider your responses to these concerns and assess a revised manuscript before we make a final decision on publication.

We therefore invite you to revise and resubmit your manuscript, along with a point-by-point response that takes into account the points raised. Please highlight all changes in the manuscript text file.

Please submit your point-by-point responses as a separate file, distinct from your cover letter where you can add responses to the Editors' comments that you do not want to be made available to the reviewers. Word files are preferred. We recommend that any figures, tables or graphs that are included in the response to reviewers are also included in the main article or Supplementary Information.

Please use the following link to submit your revised manuscript, point-by-point response to the referees' comments (which should be in a separate document to any cover letter), a tracked-changes version of the manuscript (as a PDF file) and the completed checklist:

Link Redacted

We hope to receive your revised paper within six weeks; please let us know if you aren't able to submit it within this time so that we can discuss how best to proceed. If we don't hear from you, and the revision process takes significantly longer, we may close your file. In this event, we will still be happy to reconsider your paper at a later date, as long as nothing similar has been accepted for publication at Communications Earth & Environment or published elsewhere in the meantime.

Please do not hesitate to contact us if you have any questions or would like to discuss these revisions further. We look forward to seeing the revised manuscript and thank you for the opportunity to review your work.

Best regards,

Sylvia Sullivan, PhD
Editorial Board Member
Communications Earth & Environment

Alireza Bahadori, PhD
Associate Editor
Communications Earth & Environment

EDITORIAL POLICIES AND FORMATTING

Editorial Policy: [Policy requirements](https://www.nature.com/documents/nr-editorial-policy-checklist.pdf) (Download the link to your computer as a PDF.)

- Behavioural and social science
- Ecological, evolutionary & environmental sciences
- Life sciences

<https://www.nature.com/documents/nr-reporting-summary.zip>

Furthermore, please align your manuscript with our format requirements, which are summarized on the following checklist: [Communications Earth & Environment formatting checklist](https://www.nature.com/documents/commsj-phys-style-formatting-checklist-article.pdf)

and also in our style and formatting guide [Communications Earth & Environment formatting guide](https://www.nature.com/documents/commsj-phys-style-formatting-guide-accept.pdf) .

*** DATA: Communications Earth & Environment endorses the principles of the Enabling FAIR data project (<http://www.copdess.org/enabling-fair-data-project/>). We ask authors to make the data that support their conclusions available in permanent, publically accessible data repositories. (Please contact the editor if you are unable to make your data available).

All Communications Earth & Environment manuscripts must include a section titled "Data Availability" at the end of the Methods section or main text (if no Methods). More information on this policy, is available at <http://www.nature.com/authors/policies/data/data-availability-statements-data-citations.pdf>.

If a community resource is unavailable, data can be submitted to generalist repositories such as [figshare](https://figshare.com/) or [Dryad Digital Repository](http://datadryad.org/). Please provide a unique identifier for the data (for example a DOI or a permanent URL) in the data availability statement, if possible. If the repository does not provide identifiers, we encourage authors to supply the search terms that will return the data. For data that have been obtained from publically available sources, please provide a URL and the specific data product name in the data availability statement. Data with a DOI should be further cited in the methods reference section.

REVIEWER COMMENTS:

Reviewer #1 (Remarks to the Author):

General comments

This study shows that the strong stratospheric polar vortex clearly contributed to the increased storminess (and thus surface impacts) in Northern Europe during the unusually stormy February 2022. Thanks to the nice used set of statistical as well as physical tools (such as the nudging experiments), this conclusion is very convincing to me. The manuscript is clearly written and most of the important method details can be found in the methods section. Overall, I thus only have a range of minor comments, and none of these comments require any substantial changes in the analysis. Nevertheless, I would like to highlight the two most important comments by listing them as major comments hereafter.

Major comments

Although you carefully mention throughout the manuscript that your conclusions hold for February 2022, I think you should discuss a bit more that they do not necessarily apply to all strong SPV phases (you could do so after the discussion at L184-187). Yes, my gut feeling also tells me that other phases of strong SPV states likely yield a similar tropospheric response, especially because a strong SPV is a relatively stable and zonally symmetric state that probably causes less case-to-case variability (in strong contrast to weak SPVs / SSWs, which differ a lot from case to case). However, I think you would have to prove this first, if the relative influence from the stratosphere is similar for “all” strong SPVs. Therefore, I think this should be one of the future-work aspects you should include at the end (I think it would be very powerful to just repeat your study for “all” strong SPVs in the past). The reason for why I say this is that I think the tropospheric state likely also played a role in how strongly the strong SPV affected the troposphere. In other words, it could have been – to some degree by chance – that the large-scale / regime evolution in February 2022 was such that it helped forming many strong jet streaks at the right place to foster strong cyclogenesis. Another, slightly different tropospheric setup in another strong-SPV winter could thus be less (or also more) “receptive” to the influence of the strong SPV. I think this aspect should be clearly mentioned in your discussion, although it does not diminish at all that you clearly show the important role during February 2022.

Particularly in your discussion, I would highlight the potential of your finding for subseasonal-to-seasonal predictability and thus early warning a bit more. I am not sure anymore if it really was February 2022 (I think so), but I remember looking at subseasonal regime forecasts during this period, and it was striking how far ahead the model was able to correctly predict them – much further than we are used to. And I’m sure the stratosphere played a role there. In my personal opinion, you could even hypothesize that strong SPVs might be a so far underestimated “window of forecast opportunity”, because, in contrast to the much more widely studied weak SPVs / SSWs, they likely have a smaller case-to-case variability and thus might be a more robust “driver of predictability” (cf. comment above). Of course these are all hypotheses, but I think it would be worth mentioning something like this in the discussion.

Minor comments

Fig. 1b: I find it confusing that the title says “storm intensity” but the caption says “storm count”. Maybe rephrase the title to something like “storm activity”?

L34: I guess the verb “were” is missing between “storms” and “tracked”

L39: What are “monthly maximum wind gusts”? Maximum values of (6-hourly?) gusts found within one month? Please specify.

L44: “in the UK and Europe” sounds strange to me, so I would rather say “in the UK and the rest of Europe” (or “in the UK and the EU”, if this is what you meant)

L53: add “altitude” after “10-50km”

Fig. S1: Could you add two dashed lines showing for instance the 10th and 90th percentiles of the ERA5 climatological wind speeds – in addition to the black line showing the ERA5 climatological mean? This would help to immediately see how extreme February 2022, i.e. the orange line, was in ERA5.

L63: Maybe you can add some other citations here (in addition to Kidston et al. 2015), such as Ambaum & Hoskins (2002, [https://doi.org/10.1175/1520-0442\(2002\)015<1969:TNTSC>2.0.CO;2](https://doi.org/10.1175/1520-0442(2002)015<1969:TNTSC>2.0.CO;2)).

L70: I think it would be good if you already mentioned the aspect of potentially enhanced predictability for cyclone clustering due to a strong SPV, which, I guess, is one of the key motivations for this study (since strong SPVs can persist for quite long). You do discuss it in the discussion, but I guess you could also mention it right here after the last sentence, which says that knowing about the relationship between SPV and cyclone clustering is important.

L82: Did you actually look at the NAO index during February 2022? Is it anomalously high? Maybe something worth mentioning in the paper (and maybe even showing it in the supplement).

L82-83: What exactly do you mean with “north-west/south-east shift of the dipole pattern”? It’s hard to imagine... Is it like stretching of it?

Fig. S2: It looks like there is a grayish filter on this figure – is it true?

Fig. 2: It would probably be good to adjust Fig. 2d to the exact same height as the panel on the left, to make sure the (same) y-axes are aligned. Furthermore, I think you should change the colors of the horizontal dashed lines. I only realized later that the green for Dudley and blue for Eunice have nothing to do with the same experiment colors. Finally, are the indicated minimum pressures of the three real storms (Dudley etc.) the pressure values these storms had in the UK box or potentially somewhere else (i.e., earlier / upstream over the Atlantic)?

L177-178: You could also add some further studies here that showed the potential of strong (i.e., not just weak) SPVs for longer-range forecast skill (although not specifically for cyclones but surface weather in general), such as Tripathi et al. (2015, <https://doi.org/10.1088/1748-9326/10/10/104007>), Beerli et al. (2017, <https://doi.org/10.1002/qj.3158>), or Büeler et al. (2020, <https://doi.org/10.1002/qj.3866>). In particular, Büeler et al. (2020) showed that for many European re-gions, initial strong SPVs tend to be a more “robust” “window of forecast opportunity” than the much more widely studied initial weak SPV states (or SSWs), particularly for Central and Southern Europe.

L195: Maybe it would be better to replace “predictions” with “early warnings” here? Because the time-scale you deal with is rather interesting for the latter, while the details of the impacts of individual storms will then be captured by medium-range weather forecasts.

L225-230: When I click on all the given links here, it says the page cannot be found. So, please check before publication.

L227-228: I would write "...we use the operational forecasts..."

L234-235: Stupid question, but how do you get a value over a 3-second period? Do these models even have this temporal resolution? Or is 3 seconds the model time step? Furthermore, are this maximum effective wind gusts stronger than the "normal" wind gusts since previous post-processing?

L234: Is the first "effective" needed?

L235: "for the duration of each time" sounds a bit strange... You mean each time step?

L253-255: Isn't this skill even on a 2-month lead time? I thought the forecasts were initialized on 1st of January, and you evaluate the whole February. Or is this a terminology issue?

Fig. S1: Have you ever considered moving this figure into the main manuscript? It's probably the figure I looked at the longest, and it's quite crucial for the whole manuscript, right?

L251-253: Is this something to be expected, that the weak vortex subset aligns with the climatological ERA5 wind? Or is it just more or less by chance that the lower end of the spread of the multi-model ensemble during that event approximately matched the climatological mean of the vortex? The way you write it here sounds a bit like this is to be expected... Adding the whole ERA5 climatological distribution (i.e., not just the mean, as suggested earlier) might additionally help to put this “weak SPV ensemble” a bit into context.

Fig. S1 (strongly related to the previous comment): I'm not sure if this question makes sense, but I thought about it quite a bit and still get confused: In your whole study, for the C3S simulations you always compare metrics for the “strong SPV members” against the “weak SPV members”, and you call the “weak SPV members” “climatology” (which is justified from the fact that indeed its mean matches with the ERA5 climatology in Fig. S1). However, what would happen if you would use the real C3S model February climatology as a reference instead of the “weak SPV members” (of course, this would require using hindcasts for all models)? Would this curve and its spread look the same as the one of your “weak SPV members” shown in green? And would this somehow change any conclusions of your study, also for instance for the C3S Figures such as S2b, S3b,e etc.?

L265: Maybe add "(i.e., at higher altitudes)" after "50 hPa", just to be clear that the main nudging is in the middle stratosphere. Also, maybe you should explicitly write in words that the nudging is strongest in the middle stratosphere but slowly tapers off in the lower stratosphere (if I understood it correctly)?

L269: Delete "to" before "both theoretically"

L268-271: I'm not an expert on nudging experiments, so I don't fully understand the point here: 2D nudging means that you nudge towards a state with the dimensions z (vertical profile) and y (each latitude), while the x dimension is missing because it's a zonal mean? And why does this approach ensure eddies to vary dynamically across the tropopause? I assume you refer to vertically propagating waves, which are crucial for the disturbance of the vortex? And what happens to horizontally propagating waves at the tropopause – and what would happen to them if you would nudge towards a full 3D field rather than a zonal mean state (is a nudging towards a zonal mean justified because a strong vortex is more or less zonally symmetric)? Furthermore, why do you talk about the eddies across the tropopause, where the nudging is not effective? Sorry, these might be stupid questions I could probably get answered in the cited papers, but maybe a little more insight for the unexperienced reader might be useful...

L299-306: These filter thresholds sound a little bit arbitrary to me, especially since no justification for them is given. Why the 2.5d lifetime threshold? I feel like this lifetime threshold is usually lower in other identification/tracking schemes. And what is the physical argument for selecting a maximum propagation distance of maximum 9°? I know there are many degrees of

freedom and there is no right or wrong for all these algorithms, but some physical justification (or also mentioning some results from sensitivity tests, in case you performed them) might be useful here. Since you apply the same thresholds to all datasets that you compare (ERA5 and model simulations), I think the exact choice of these thresholds is indeed not critical. However, numbers like in the histogram Fig. 1b might be quite sensitive to these thresholds. So, I would suggest you write a few words about this, maybe also with a reference to Neu et al. (2013, <https://doi.org/10.1175/BAMS-D-11-00154.1>), who inter-compared a set of different track-ing algorithms.

L307: I think you don't mention in the whole paragraph before, in which you explain the method, that you "first" apply the algorithm to ERA5. You only start to mention here that you apply it to hindcasts. So maybe write somewhere before that you apply the algorithm also on 23 (?) years of ERA5 (basis for Fig. S5).

L337: I know determining such a radius in an objective way is very difficult. However, 10° seems relatively large to me, from an impact point of view (if I calculated correctly, it refers to about 600km). It could mean that a cyclone center can be quite far away from the UK, with the corresponding cyclone probably having a relatively small impact (assuming, for instance, the track would be more towards the north such that it just "brushes" the UK). Did you check the sensitivity of the cyclone counts to this number?

Reviewer #2 (Remarks to the Author):

Review of manuscript submitted to Communications Earth & Environment

Manuscript identification number: COMMSENV-24-2735-T

Title: Strong polar vortex favoured intense Northern European storminess in February 2022

Authors: Ryan Williams, Amanda Maycock, Vincent Charnay, Jeff Knight, Inna Polichtchouk

Recommendation: Minor Revision

General Comments

The authors analyse the possible links between a stormy period over Western Europe in February 2022 and the strength of the wintertime stratospheric circulation in the previous weeks. While such a link has been previously been hypothesized (e.g. Rupp et al., 2022), the authors quantify based on a multi-model S2S ensemble in how far an intensified stratospheric polar vortex contributed (in a probabilistic way) to a more probable stormy period in February 2022 on the sub-seasonal time scale. I think the authors present a well done and very interesting study which is worth publishing, but several aspects must be better explain before acceptance. The list of comments can be found below. I would be willing to re-review the manuscript upon resubmission if required.

Minor Comments

a) It is not clear while reading the introduction on what are the physical mechanisms leading to serial cyclone clustering and in particular on the crucial importance of having a quasi-stationary large-scale flow for several days for this to happen. See e.g. Dacre and Pinto (2020), page 48, section "dynamics associated with serial cyclone clustering". I think this information would be very helpful to explain the hypothesis of the paper, how the occurrence of the strong stratospheric vortex may have influenced a steady, stronger than normal westerly flow over the North Atlantic and Western Europe.

b) also introduction, line 34: The definition of a clustering event was first presented in Pinto et al. (2014). Please add the reference here.

c) Discussion, lines 170ff: It would be excellent if the authors could elaborate more in how far the February 2022 might be representative from other similar situations or not. Given the strong importance of the conditions at the tropopause level and internal tropospheric variability of the downward propagation (or not) of signals from the stratosphere, I assume this may be analogue to stratospheric sudden warming and cold waves in Europe: A sudden warming enhances the probability of a cold wave in the following weeks (e.g. Kautz et al., 2020), but it is neither a necessary nor a sufficient condition.

d) Methods, cyclone tracking, line 290ff: It would be important to add some information on how the cyclone statistics produced by this method compared to others (see Neu et al., 2013). Note that Pinto et al. (2016) provided evidence that measures of clustering can be very sensitive to the absolute number of cyclones tracks identified by each tracking method, and thus lead to quite different statistics.

e) Methods, SSI, lines 360ff: I cannot follow the justification for the use of a 95. Percentile instead of the commonly used 98. Percentile. In order to justify such a step, the authors would actually need to provide evidence that significant building losses occur already if the 95. Percentile is exceeded in the study and that the loss grows with v^3 above this value. As far as I know, there is no evidence for the above. Please enhance.

References:

Dacre and Pinto (2020) doi:10.1038/s41612-020-00152-9

Kautz et al. (2020) doi:10.1002/qj.3724

Neu et al. (2013) doi:10.1175/BAMS-D-11-00154.1

Pinto et al. (2014) doi:10.1002/2014JD022305

Pinto et al. (2016) doi:10.3402/tellusa.v68.32204

Rupp et al. (2022) doi:10.1002/2021JD035667

Reviewer #3 (Remarks to the Author):

Comments on "Strong polar vortex favoured intense Northern European storminess in February 2022" by Williams et al.

Summary

Using the ERA5 reanalysis and the C3S datasets, and the experiments to explore the possible impact of strong stratospheric polar vortex on the storminess in northern Eurasia. Most of the key findings are shown in the supplementary materials, and the result parts are too simple and short. Further, there is still a debate or a concern about the relationship between the stratospheric polar vortex and the storm on the Atlantic-Eurasia sector. Study on the possible impact of the polar vortex on the storm is not consistent between this study and others. It confuses readers whether the strong or the weak polar vortex increases the possibility of the gust and rainfall in Eurasia. Due to those confusions, I suggest a major revision at the present time.

Major comments

1. Most of the key findings are shown in the supplementary materials, and the discussion on the supplementary takes up most of the draft than the existing results concerned Figures 1-4. Key findings should be shown in the main body of the study rather than shown in the supplementary.
2. Some of the previous studies in the introduction link the weak polar vortex and the European storm, while this study links the strong polar vortex and the storm. I am wondering which is true. This study finds that strong polar vortex is more zonally uniform than the weak polar vortex. How does the round strong polar vortex affect the European rainfall and wind gust if not displaced from the Arctic?
3. I can understand the difference between strong and weak polar vortex for C3S models. However, how was the weak polar vortex selected from the model datasets? The weak polar vortex is also misleading due to the fact that the weak polar vortex wind from the C3S models is nearly identical to the climatology wind from ERA5. I do not understand why the weak polar vortex shows the wind identical to the climatology.
4. This study states that the strong polar vortex has a strong impact on the storminess. However, the difference between the strong polar vortex and the climatology or the difference between the OBS experiment and the CLIM experiments is not tested. I wonder the significance level of the strong polar vortex impact on the storm.
5. Another aspect is the physical mechanism between the strong polar vortex and the surface gust. How can we understand the relation between the strong polar vortex and the wind at surface? This study assesses the wind speed difference between the strong and "weak" polar vortex. Consistency of wind direction is not shown.

Other comments

1. L23-25: This is a case study, and the generality of this study is still not known. Since most studies reported the possible link between the weak polar vortex and the extremes in Eurasia, I am still not sure if this case can stand for all strong polar vortex events.
2. L34-35: The 4th is not a reasonable excuse to study this case. I am interested to know if the strong polar vortex is also accompanied with the extremes for the other three cases.
3. L38: What is the state of the Arctic sea ice and the tropical SSTs?
4. L67-68: I still do not follow how the strong polar vortex affects the storminess.
5. L84: Positive NAO corresponds to less storms, rather than more storms in midlatitudes.
6. L121: Not exactly UK, but a wider region.
7. L130: How does the strong westerly jet promote cyclogenesis?
8. L151: Is there a possibility that the increase in the wind speed is due to the sample bias? I have no idea to what extent this increase is true.
9. L188: A statistic is possible if you examine the historical records using the ERA5 reanalysis.
10. Table 1: Where did you show the C3S results? Are the daily data available for the seven seasonal models? We do know which initializations are available and shown.
11. L259: How can we be sure that the data in other members are not problematic?
12. L265-268: Are the nudging only done between 50-90 hPa? But you define polar vortex at 100 hPa?

13. L273: This is an ill sentence.
14. Figure 1: b, what timespan is this histogram is based on? c. monthly maximum is too occasional to explore the impact of strong vortex on wind speed. What about the monthly mean?
15. Figure 2: Weak and strong polar vortex are too subjective to define here. Methods for d is missing. Please clarify.
16. Figure 3: Results are not consistent between C3S models and experiments.
17. Figure S1: It is weird to see the OBS climatology is equal to weak in a.
18. Figure S2: Monthly or daily data used here?

Communications Earth & Environment is committed to improving transparency in authorship. As part of our efforts in this direction, we are now requesting that all authors identified as 'corresponding author' create and link their Open Researcher and Contributor Identifier (ORCID) with their account on the Manuscript Tracking System prior to acceptance. ORCID helps the scientific community achieve unambiguous attribution of all scholarly contributions. You can create and link your ORCID from the home page of the Manuscript Tracking System by clicking on 'Modify my Springer Nature account' and following the instructions in the link below. Please also inform all co-authors that they can add their ORCIDs to their accounts and that they must do so prior to acceptance.

Version 1:

Decision Letter:

Dear Dr Williams,

Your revised manuscript titled "Strong polar vortex favoured intense Northern European storminess in February 2022" has now been seen by our reviewers, whose comments appear below. In light of their advice, we are delighted to say that we are happy, in principle, to publish a suitably revised version in Communications Earth & Environment, provided you address the outstanding concern of Reviewer 3 about correspondence of the weak polar vortex and vortex climatology. If there are other existing studies on the February 2020 SPV or others and their effects on storminess, we request that these be added to the Introduction also, per Reviewer 3's major comment 2.

We therefore invite you to revise your paper one last time to address the remaining concerns of our reviewers. At the same time we ask that you edit your manuscript to comply with our format requirements and to maximise the accessibility and therefore the impact of your work.

EDITORIAL REQUESTS:

****Please take care to match our formatting and policy requirements. We will check revised manuscript and return manuscripts that do not comply. Such requests will lead to delays. ****

SUBMISSION INFORMATION:

OPEN ACCESS:

Communications Earth & Environment is a fully open access journal. Articles are made freely accessible on publication. For further information about article processing charges, open access funding, and advice and support from Nature Research, please visit <https://www.nature.com/commsenv/open-access>

Link Redacted

Best regards,

Alireza Bahadori, PhD
Associate Editor
Communications Earth & Environment

Sylvia Sullivan, PhD
Editorial Board Member
Communications Earth & Environment

REVIEWERS' COMMENTS:

Reviewer #1 (Remarks to the Author):

I would like to thank the authors for addressing all the reviewer comments so thoroughly! I went through the response document as well as the revised manuscript, and I think the revision improved it quite a bit, particularly due the inclusion of some (new or modified) figures into the main manuscript. Therefore, I have no further comments and suggest the manuscript to be published. Congratulations for this nice work!

Reviewer #2 (Remarks to the Author):

Dear authors, I have happy with the replies and the revised version of the manuscript and thus suggest its acceptance for publication.

Reviewer #3 (Remarks to the Author):

Comments on "Strong polar vortex favoured intense Northern European storminess in February 2022" by Williams et al.

Summary

The largest concerns in my previous review were not seriously considered by the authors. Specifically, the weak polar vortex defined in this study is very different from other studies in literature. The weak polar vortex and the vortex climatology is identical in this study. Therefore, I did not see any scientific value added to existing literature. Further, the study is focused on a very narrow area, which might not attract wide attention to this study. After careful consideration, I do not recommend publication of this study.

Major comments

1. This study states that both weak and strong polar vortex can cause wide storm activities in Europe. If so, I can not link the storminess to the vortex, because the storm activities do not depend on the polar vortex. If a polar vortex is strong over the Arctic, the weather in Europe is relatively tranquil. However, the authors argue that both strong and weak polar vortex

explain the storminess in Europe. I am seriously doubting about those results.

2. This is a case study, and a comparison between this case and others is necessary. For example, if the findings in this study is also applied to other cases. The strong polar vortex was also observed in 2020, a review on previous studies is necessary to further confirm finding in this study. I find results in this study might be not consistent with other reported. For example, the strong polar vortex in 2020 was accompanied with an ozone loss over the Arctic and warmer spring. Please refer to several studies here for reference (<https://doi.org/10.1029/2020JD034190>; <https://doi.org/10.1029/2020JD033524>). Although the month focused is a bit different, a review and discussion is required to confirm the consistency or inconsistency between this study and others. The review in this study is too narrow, and a wide review should be performed.

3. The structure of this study is also problematic, and key results are not placed in the main body of this study, but in supplementary. The authors argue that the word limited is 5000 words. This limit can include most of figures in the supplementary, and the 5000 word limit should exclude reference lists. Therefore, I am not satisfactory with the revision of this study, although comments from other two reviewers were seriously considered. My comments were not.

Specific comments

1. The response “We do not agree that our results are inconsistent with earlier studies on either weak or strong polar vortex influence on European climate...” Consistency with Afargan-Gerstman et al., 2024 does not ensure consistency with other literature. Therefore, my comments should be considered again. What aspect is consistent, and what aspect is inconsistent? Please make a careful review.

2. The reply “As explained in the methods (L317-326), the weak members from the C3S models were chosen as those with extratropical lower stratospheric February zonal winds below the 20th percentile of the ensemble members.” I did not agree with this method to choose weak and strong polar vortex. The strong and weak polar vortex should be selected from a complete continuous dataset. For ideal forecasts, the polar vortex should be consistently strong or weak in models. That is, you can not choose weak polar vortex from an ensemble of strong polar vortex. You also can not choose strong polar vortex from an ensemble of weak polar vortex. This method is not reasonable and scientific. To solve this problem, you should use all hindcasts at least 20 or 30 years. The model climatology should be assessed first.

3. The reply “We do not understand the comment about wind direction as it does not refer to a specific figure.” If the wind direction is not considered, and the wind direction consistency is not examined, how can you ensure all the response is from the stratospheric polar vortex. This reply is unsatisfactory for reviewer.

4. The reply “L229-238: While our results are specific to February 2022, it is plausible that other years with strong SPV conditions could exhibit a similar influence on the North Atlantic storm track.” Are you sure that your result is consistent with the 2020 case. I do not think so.

5. The reply “We do not claim that a strong SPV would explain all years with high cyclogenesis.” So what is the scientific/practical significance of this study?

6. The reply “These are initialised in the forecasts and evolve similarly between the two experiments so are unlikely to contribute to any differences we are interpreting to be the influence from the SPV. 2022 was a La Niña winter, which could have been a driver of the strong SPV as noted on L67. Arctic Sea ice extent was 2% below the February average for 1991-2020 <https://climate.copernicus.eu/sea-ice-cover-february-2022>.” I do not agree with this. The model difference might be atmospheric chaos or unstable model error. The oceanic forcing might be important.

7. The comment in previous review “5. L84: Positive NAO corresponds to less storms, rather than more storms in midlatitudes.” If this study focuses on the high latitudes or even Arctic, the scientific significance is even weak. Since the Arctic is covered with the polar vortex most of the time, and it should be so (stormy) most of the time. It is not unexpected any more. I did not see any special value of this study.

8. The reply “This is beyond the scope of the current work and will be a topic of future study.” I do not think it is beyond the scope of this study.

9. Structure of the paper is disordered. Table 1 and Table 2 should be in order. My comment was not answered directly: “10. Table 1: Where did you show the C3S results? Are the daily data available for the seven seasonal models? We do know which initializations are available and shown.”

So I ask once again: Are the daily data available for the seven seasonal models? We do know which initializations are available and shown.

10. The reply “We think the reviewer has not interpreted the legend clearly. The C3S, GloSea6 and IFS members are combined here into two groups: one representing weak SPV and one representing strong SPV. Therefore, there is no problem with consistency.” Where is the updated figure? Where are the weak SPV shown in the revision. Please tell me where you show and what revision did you do? So this comment was overlooked again?

Reviewer #1 (Remarks to the Author):

General comments

This study shows that the strong stratospheric polar vortex clearly contributed to the increased storminess (and thus surface impacts) in Northern Europe during the unusually stormy February 2022. Thanks to the nice used set of statistical as well as physical tools (such as the nudging experiments), this conclusion is very convincing to me. The manuscript is clearly written and most of the important method details can be found in the methods section. Overall, I thus only have a range of minor comments, and none of these comments require any substantial changes in the analysis. Nevertheless, I would like to highlight the two most important comments by listing them as major comments hereafter.

Thank you for reviewing our manuscript. We are pleased that our approach can easily be followed and that the conclusions are convincing. We are grateful for your constructive comments and hope that our responses address your minor comments and further enhance the clarity of the manuscript.

Major comments

Although you carefully mention throughout the manuscript that your conclusions hold for February 2022, I think you should discuss a bit more that they do not necessarily apply to all strong SPV phases (you could do so after the discussion at L184-187). Yes, my gut feeling also tells me that other phases of strong SPV states likely yield a similar tropospheric response, especially because a strong SPV is a relatively stable and zonally symmetric state that probably causes less case-to-case variability (in strong contrast to weak SPVs / SSWs, which differ a lot from case to case). However, I think you would have to prove this first, if the relative influence from the stratosphere is similar for “all” strong SPVs. Therefore, I think this should be one of the future-work aspects you should include at the end (I think it would be very powerful to just repeat your study for “all” strong SPVs in the past). The reason for why I say this is that I think the tropospheric state likely also played a role in how strongly the strong SPV affected the troposphere. In other words, it could have been – to some degree by chance – that the large-scale / regime evolution in February 2022 was such that it helped forming many strong jet streaks at the right place to foster strong cyclogenesis. Another, slightly different tropospheric setup in another strong-SPV winter could thus be less (or also more) “receptive” to the influence of the strong SPV. I think this aspect should be clearly mentioned in your discussion, although it does not diminish at all that you clearly show the important role during February 2022.

Particularly in your discussion, I would highlight the potential of your finding for subseasonal-to-seasonal predictability and thus early warning a bit more. I am not sure anymore if it really was February 2022 (I think so), but I remember looking at subseasonal regime forecasts during this period, and it was striking how far ahead the model was able to correctly predict them – much further than we are used to. And I’m sure the stratosphere played a role there. In my personal opinion, you could even hypothesize that strong SPVs

might be a so far underestimated “window of forecast opportunity”, because, in contrast to the much more widely studied weak SPVs / SSWs, they likely have a smaller case-to-case variability and thus might be a more robust “driver of predictability” (cf. comment above). Of course these are all hypotheses, but I think it would be worth mentioning something like this in the discussion.

Thanks for the comment. We agree we need to carefully caveat that our conclusions are specific to the February 2022 case study. We have amended the discussion to reflect this. We agree it would be valuable to apply our approach to other strong SPV events to determine if the influence in February 2022 was typical or atypical; we hope this study will motivate the community to do this.

The discussion now reads (L229-338): “While our results are specific to February 2022, it is plausible that other years with strong SPV conditions could exhibit a similar influence on the North Atlantic storm track. For example, at the time of writing, February 2020 had the strongest monthly SPV since 1979 (Lawrence et al., 2020) and also exhibited intense North Atlantic cyclogenesis (Davies et al., 2021). However, the influence of the SPV on the storm track in February 2022 may have been shaped by other factors related to the tropospheric state at the time, which are captured in the initialised model experiments. It would be insightful to apply the attribution approaches used here to study other periods with strong SPV conditions, to complement composite approaches (Afargan-Gerstman et al., 2024).”

And we mention the potential for windows of opportunity in S2S prediction (L216-224): “While it was not the focus of this study, it is notable that the C3S models initialised on 1 January 2022 were confident the February SPV would be stronger than average; this is evident from Fig. 2a where the distribution of weak C3S ensemble members is clearly shifted to higher values than the observed climatology. The signal for a strong SPV was evident from forecasts initialised as early as November 2021 using GloSea6 (McLean et al., 2024). This suggests that strong SPV states could offer a ‘window of opportunity’ for enhanced European predictability, particularly over Central and Southern Europe (Büeler et al., 2020).”

Minor comments

Fig. 1b: I find it confusing that the title says “storm intensity” but the caption says “storm count”. Maybe rephrase the title to something like “storm activity”?

Thanks for spotting this error. We have corrected this in the revised manuscript.

L34: I guess the verb “were” is missing between “storms” and “tracked”

This has been added.

L39: What are “monthly maximum wind gusts”? Maximum values of (6-hourly?) gusts found within one month? Please specify.

The quantity we evaluate across all the model simulations (including ERA5) is the maximum 10 m wind gust since previous post-processing. It is defined here as the ‘maximum 3 second wind at 10 m height as defined by the WMO’: <https://codes.ecmwf.int/grib/param-db/49>

For a given timestep (e.g., 6-hourly), the highest 3-second gust is output and we then take the maximum of all 6-hourly maxima. This yields the maximum sustained 3-second gust during February 2022. This is explained in the Methods.

L44: “in the UK and Europe” sounds strange to me, so I would rather say “in the UK and the rest of Europe” (or “in the UK and the EU”, if this is what you meant)

We have added ‘the rest of’ between ‘UK’ and ‘Europe’.

L53: add “altitude” after “10-50km”

Done

Fig. S1: Could you add two dashed lines showing for instance the 10th and 90th percentiles of the ERA5 climatological wind speeds – in addition to the black line showing the ERA5 climatological mean? This would help to immediately see how extreme February 2022, i.e. the orange line, was in ERA5.

This is a great suggestion. Since we bring this Figure into the main manuscript now (Fig. 2), we first combined the result for both GloSea6 and IFS for brevity (the signal is roughly approximate), labelling this as ‘SNAPSI’ (Fig. 2b). Then, this information from ERA5 is instead included as a boxplot (Fig. 2c) to demonstrate the exceptional nature of the SPV strength during February 2022 (in excess of the 90th percentile and second only to 2020 by a very fine margin).

L63: Maybe you can add some other citations here (in addition to Kidston et al. 2015), such as Ambaum & Hoskins (2002, [https://doi.org/10.1175/1520-0442\(2002\)015<1969:TNTSC>2.0.CO;2](https://doi.org/10.1175/1520-0442(2002)015<1969:TNTSC>2.0.CO;2)).

We have added this citation/reference alongside Kidston et al. (2015):

Ambaum, M. H., & Hoskins, B. J. (2002). The NAO troposphere–stratosphere connection. *Journal of Climate*, 15(14), 1969-1978.

L70: I think it would be good if you already mentioned the aspect of potentially enhanced predictability for cyclone clustering due to a strong SPV, which, I guess, is one of the key motivations for this study (since strong SPVs can persist for quite long). You do discuss it in the discussion, but I guess you could also mention it right here after the last sentence, which says that knowing about the relationship between SPV and cyclone clustering is important.

This is an additional point which we ought to have mentioned here and we thank you for highlighting this. We have now embedded the following couple of additional sentences within this paragraph (L75-80): “Given that strong SPV conditions often persist for several weeks or longer, this could offer a source of subseasonal predictability for the position of the

storm track and main regions of cyclogenesis (Afargan-Gerstman et al., 2024; Rupp et al., 2024). The influence of a strong SPV on the jet stream could also affect serial cyclone clustering, which has been linked to a persistent, zonally orientated and intensified jet over the eastern North Atlantic (Pinto et al., 2014).”

L82: Did you actually look at the NAO index during February 2022? Is it anomalously high? Maybe something worth mentioning in the paper (and maybe even showing it in the supplement).

Thanks for this great suggestion. We have calculated the NAO index anomaly for ERA5 (versus 1979-2021 climatology), C3S and the SNAPSI experiments (GloSea6 and IFS), with values computed as +2.05, +1.48 and + 1.74 (+1.78 and +1.74) respectively. These values are overlaid onto new Fig. 3d-f and Fig. S2 accordingly. The methodology of calculating this has been added to the Methods (L381-401).

Note that the new Fig. 3 combines the results for the jet stream, MSLP and storm track anomaly signal (note GloSea6 and IFS results are combined together as ‘SNAPSI’). This decision has been made following particular reviewer 3’s suggestion of including more results into the main manuscript.

The following has been added to reflect the insertion of more results into the main manuscript, particularly in addressing reviewer 3’s concern about this, in addition to the mentioning the NAO signal we’ve since calculated:

L99-102: “The strong SPV in February 2022 induces a poleward shifted, intensified North Atlantic jet stream (Fig. 3a-c; Fig. S1), a MSLP anomaly pattern that resembles a positive NAO index (Fig. 3d-f; Fig. S2), and an enhanced and poleward shifted storm track from Newfoundland to the Norwegian Sea (Fig. 3-g-i; see also Fig. S3).”

L104-107: “In computing the NAO index anomaly associated with the strong SPV signal according to the models, we estimate an ensemble mean value of $\sim+1.5-1.8$ which is close to that according to the reanalysis (+2.05; see Methods), which is the 6th most positive value for February since 1979.”

L82-83: What exactly do you mean with "north-west/south-east shift of the dipole pattern"? It's hard to imagine... Is it like stretching of it?

The orientation of the NAO-like MSLP dipole anomalies according to ERA5 is largely zonal across the North Atlantic. Whereas in the SPV strength experiments, the orientation of the MSLP dipole between higher and lower latitudes is slightly rotated anticlockwise, as found by earlier studies in connection with anomalous vortex states (e.g., Knight et al., 2021).

We decided on removing this sentence as it may detract the reader from following the main story as it is potentially hard to visualise.

Fig. S2: It looks like there is a grayish filter on this figure – is it true?

We tried this because we wanted to more clearly display multiple layers of information overlapped, but we have removed it as this made little difference. The same applies to old Fig. S7 (now Fig. 3a-c and Fig. S1).

Fig. 2: It would probably be good to adjust Fig. 2d to the exact same height as the panel on the left, to make sure the (same) y-axes are aligned. Furthermore, I think you should change the colors of the horizontal dashed lines. I only realized later that the green for Dudley and blue for Eunice have nothing to do with the same experiment colors. Finally, are the indicated minimum pressures of the three real storms (Dudley etc.) the pressure values these storms had in the UK box or potentially somewhere else (i.e., earlier / upstream over the Atlantic)?

Agreed. We had produced Figure 2d separately but now merge all panels into one plot (now Figure 4), so the vertical axes line up precisely.

We have changed the line colours for the named storms to avoid confusion with the experiment labels (also Fig. S6). We choose colours that are distinct but different to those used in highlighting results for the reanalysis and the models.

The pressure values denote the maximum intensity over the UK region for each named storm. We have amended the Figure caption to emphasise this more clearly. We have also added lines for the minimum lifetime pressure for each named storm within the entire North Atlantic domain Fig. S6 for comparison.

L177-178: You could also add some further studies here that showed the potential of strong (i.e., not just weak) SPVs for longer-range forecast skill (although not specifically for cyclones but surface weather in general), such as Tripathi et al. (2015, <https://doi.org/10.1088/1748-9326/10/10/104007>), Beerli et al. (2017, <https://doi.org/10.1002/qj.3158>), or Büeler et al. (2020, <https://doi.org/10.1002/qj.3866>). In particular, Büeler et al. (2020) showed that for many European re-gions, initial strong SPVs tend to be a more “robust” “window of forecast opportunity” than the much more widely studied initial weak SPV states (or SSWs), particularly for Central and Southern Europe.

Thank you for pointing us to these papers. We now cite both Tripathi et al. (2015) and Beerli et al. (2017) at the end of this sentence (L215-216). The Büeler et al. (2020) paper we cite afterwards (L224) in noting the disparity in enhanced predictability for Central and Southern Europe following strong versus weak/SSW SPV states.

L195: Maybe it would be better to replace “predictions” with “early warnings” here? Because the time-scale you deal with is rather interesting for the latter, while the details of the impacts of individual storms will then be captured by medium-range weather forecasts.

We agree that ‘early warnings’ is more appropriate than ‘predictions’ in this context. We have made this change.

L225-230: When I click on all the given links here, it says the page cannot be found. So, please check before publication.

Thanks for pointing this out. These were affected by the transition of the CDS to a new service. The links have been updated.

L227-228: I would write "...we use the operational forecasts..."

Have added 'operational' before forecasts here.

L234-235: Stupid question, but how do you get a value over a 3-second period? Do these models even have this temporal resolution? Or is 3 seconds the model time step? Furthermore, are this maximum effective wind gusts stronger than the "normal" wind gusts since previous post-processing?

See reply to your earlier L39 comment. The models use a parametrization to derive an effective 3 second gust based on the instantaneous 10m wind speed at the atmospheric model timestep (usually of order 10-15 minutes). See <https://codes.ecmwf.int/grib/param-db/49>

L234: Is the first "effective" needed?

We have removed the first 'effective' which we inadvertently mentioned twice here.

L235: "for the duration of each time" sounds a bit strange... You mean each time step?

Yes indeed, we should have been more specific. Corrected.

L253-255: Isn't this skill even on a 2-month lead time? I thought the forecasts were initialized on 1st of January, and you evaluate the whole February. Or is this a terminology issue?

As we refer to the whole of February, we think it's better to say 1- to 2-month lead time for forecast initialised on 1 January. We have changed this.

Fig. S1: Have you ever considered moving this figure into the main manuscript? It's probably the figure I looked at the longest, and it's quite crucial for the whole manuscript, right?

Thanks for the suggestion. We have moved the figure to the main manuscript and include this as Figure 2 (existing earlier Figures are now renamed Figure 4-6 accordingly to accommodate the inclusion of this new Figure and the new Figure 3).

L251-253: Is this something to be expected, that the weak vortex subset aligns with the climatological ERA5 wind? Or is it just more or less by chance that the lower end of the spread of the multi-model ensemble during that event approximately matched the climatological mean of the vortex? The way you write it here sounds a bit like this is to be expected... Adding the whole ERA5 climatological distribution (i.e., not just the mean, as suggested earlier) might additionally help to put this "weak SPV ensemble" a bit into context.

This is not to be expected a priori, but is a notable feature of the ensembles because it means that the models had high confidence the SPV would be above average strength in

February 2022 at 1-2 months lead time. We now modify this sentence to make it clear that this was not expected a priori, but gives information about the SPV predictability.

We have added information of the 10th and 90th percentiles according to ERA5 monthly mean climatology (Fig. 2c) for additional context and note that the median of the strong members (Obs members) exceeds the 90th percentile consistently throughout the month (Fig. 2a).

Fig. S1 (strongly related to the previous comment): I'm not sure if this question makes sense, but I thought about it quite a bit and still get confused: In your whole study, for the C3S simulations you always compare metrics for the "strong SPV members" against the "weak SPV members", and you call the "weak SPV members" "climatology" (which is justified from the fact that indeed its mean matches with the ERA5 climatology in Fig. S1). However, what would happen if you would use the real C3S model February climatology as a reference instead of the "weak SPV members" (of course, this would require using hindcasts for all models)? Would this curve and its spread look the same as the one of your "weak SPV members" shown in green? And would this somehow change any conclusions of your study, also for instance for the C3S Figures such as S2b, S3b,e etc.?

The answer to this question depends on the C3S model biases in SPV strength. If the models are unbiased then the climatology of the full hindcast period would be close to ERA5 and the signal with regard to the Feb 2022 strong SPV should be similar to first order. However, subseasonal-to-seasonal forecast models tend to have too strong/cold SPVs (Lawrence et al., 2022) meaning the climatology of the full hindcast period would have a larger difference in SPV strength when compared with the Feb 2022 'strong' members. Therefore, this choice would be a less useful comparison because it would overestimate the signal from the SPV in Feb 2022. In contrast, the strong vs. weak C3S members for Feb 2022 have a comparable difference in SPV strength to the observed SPV vs. climatology, which is a better comparison with the nudged experiments.

Lawrence et al., Quantifying stratospheric biases and identifying their potential sources in subseasonal forecast systems, *Weather Clim. Dynam.*, 3, 977–1001, <https://doi.org/10.5194/wcd-3-977-2022>, 2022.

L265: Maybe add "(i.e., at higher altitudes)" after "50 hPa", just to be clear that the main nudging is in the middle stratosphere. Also, maybe you should explicitly write in words that the nudging is strongest in the middle stratosphere but slowly tapers off in the lower stratosphere (if I understood it correctly)?

Thanks for these suggestions, which we have added.

L269: Delete "to" before "both theoretically"

Done

L268-271: I'm not an expert on nudging experiments, so I don't fully understand the point here: 2D nudging means that you nudge towards a state with the dimensions z (vertical profile) and y (each latitude), while the x dimension is missing because it's a zonal mean?

And why does this approach ensure eddies to vary dynamically across the tropopause? I assume you refer to vertically propagating waves, which are crucial for the disturbance of the vortex? And what happens to horizontally propagating waves at the tropopause – and what would happen to them if you would nudge towards a full 3D field rather than a zonal mean state (is a nudging towards a zonal mean justified because a strong vortex is more or less zonally symmetric)? Furthermore, why do you talk about the eddies across the tropopause, where the nudging is not effective? Sorry, these might be stupid questions I could probably get answered in the cited papers, but maybe a little more insight for the unexperienced reader might be useful...

Stratospheric nudging has previously been applied in both zonal mean and 3-D configurations. Indeed, the SNAPSI experiment protocol includes both configurations, but prioritises zonal mean nudging. We chose zonal mean nudging because of the relatively high degree of zonal symmetry in strong SPV cases, certainly when compared to the onset of weak SPV cases when the vortex splits or is displaced.

Zonal mean nudging allows vertically and horizontally propagating waves to continue to propagate but on a different zonal mean state set by the nudging. This is why the vertical tapering of the nudging strength is important to avoid any sharp gradients that might cause instabilities for propagating waves. Hitchcock and Haynes (2014) show that zonal mean nudging effectively reproduces the anomalous meridional mass circulation in the underlying troposphere and Hitchcock and Simpson (2014) showed this corresponds to similar tropospheric Coriolis acceleration, in a simulation of a stratospheric sudden warming. In our case study, stratospheric planetary wave activity is suppressed during the strong SPV and so the fact that the nudged model will not reproduce identical zonal asymmetries to those observed is arguably less important. If 3-D nudging is used, then within the nudging region the wave field is effectively prescribed and any vertically or horizontally propagating waves are damped once they reach the nudging region.

We feel this is too much detail to include in the Methods but we have added a sentence (L339-342): “While nudging can be applied in 3-dimensions (e.g., Hitchcock et al., 2022), the SPV tends to be relatively zonally symmetric when it is strong, so this should be suitable to capture most of the signal, consistent with earlier studies (Douville, 2009; Hitchcock and Simpson, 2014; Zhang et al., 2018; Jiménez-Esteve and Domeisen, 2020).”

L299-306: These filter thresholds sound a little bit arbitrary to me, especially since no justification for them is given. Why the 2.5d lifetime threshold? I feel like this lifetime threshold is usually lower in other identification/tracking schemes. And what is the physical argument for selecting a maximum propagation distance of maximum 9°? I know there are many degrees of freedom and there is no right or wrong for all these algorithms, but some physical justification (or also mentioning some results from sensitivity tests, in case you performed them) might be useful here. Since you apply the same thresholds to all datasets that you compare (ERA5 and model simulations), I think the exact choice of these thresholds is indeed not critical. However, numbers like in the histogram Fig. 1b might be quite sensitive to these thresholds. So, I would suggest you write a few words about this, maybe also with a

reference to Neu et al. (2013, <https://doi.org/10.1175/BAMS-D-11-00154.1>), who inter-compared a set of different track-ing algorithms.

Thanks for your comment. We chose the Tempest Extremes algorithm because it is open access, fully documented (Ullrich et al., 2021) and is supported by the developers. We use the parameter choices outlined in Ullrich et al. (2021) for ETC tracking, which were shown to produce a reasonable climatology for the North Atlantic storm track density (their Fig 4). There is one exception, which is that we increased the 'range' parameter from 6° to 9° - the maximum distance (in ° great circle distance) that a cyclone can move between subsequent detections. This was increased because storm Eunice moved very quickly and after testing it was found that Eunice's track in ERA5 got split in two if 6° was used, so we increased this for our specific case study. We recognise different tracking algorithms can yield different results. We have added a sentence to the methods with this caveat and cited Neu et al. (2013).

L417-421: "These parameter choices are the same as those selected by Ullrich et al. (2021) except that the maximum propagation distance of 9° great circle distance over a 6-hour period is 50% larger than their study. This was altered because storm Eunice travelled very quickly, and it was found that a lower threshold caused Eunice to split into two separate tracks".

L423-426: "We note that a variety of methods exist for tracking extratropical cyclones which can produce different results (Neu et al., 2013), including in relation to cyclone clustering (Pinto et al., 2016). However, we use the same method applied to all datasets so the data are directly comparable".

L307: I think you don't mention in the whole paragraph before, in which you explain the method, that you "first" apply the algorithm to ERA5. You only start to mention here that you apply it to hindcasts. So maybe write somewhere before that you apply the algorithm also on 23 (?) years of ERA5 (basis for Fig. S5).

Thank you for your comment. We indeed appear to have neglected specific mention that we applied the tracking algorithm to the ERA5 reanalysis (restricted to 1994-2016 for the Fig. S5 evaluation to be consistent with the hindcasts). We have added a couple of sentences to this second paragraph under the 'Extratropical Cyclone Tracking' section of the methods.

L427-436: "As a first step, we performed tracking of extratropical cyclones in the ERA5 reanalysis between 1979 and 2022 output on a 1° x 1° grid. A comparison of the minimum MSLP along cyclone tracks during February in GloSea6, IFS and ERA5 are shown in Fig. S5 for the common period 1994-2016."

L337: I know determining such a radius in an objective way is very difficult. However, 10° seems relatively large to me, from an impact point of view (if I calculated correctly, it refers to about 600km). It could mean that a cyclone center can be quite far away from the UK, with the corresponding cyclone probably having a relatively small impact (assuming, for instance, the track would be more towards the north such that it just "brushes" the UK). Did you check the sensitivity of the cyclone counts to this number?

Thanks for your comment and it is a valid point. Ultimately, we needed a simple but sensible way to estimate 'potentially impactful UK storms' from the large samples of cyclones we track. Other literature producing composite cyclone structures has shown that precipitation features, and excess cyclone relative winds can extend up to 10 degrees from the cyclone centre. For example, Priestley and Catto (2022) show that closed MSLP contours and anomalous lower tropospheric wind speeds extend 10 deg from the cyclone centre. In their ETC Atlas, Dacre et al. (2012) used a 20 deg radius to composite cyclone structure. Precipitation features (fronts) tend to be slightly less extensive than wind; Hawcroft et al. (2012) used a 5 deg radius to estimate ETC related precipitation. Sinclair et al. (2020) use a 12 deg radius for cyclone composites. Based on these various studies, we chose 10 deg as a 'middle ground' distance over which the cyclones may affect surface conditions. In practice the hazards are not symmetric about the cyclone centre. However, it felt overly complicated to introduce a different radius for cyclones tracking to the south and north of the UK, so we applied a symmetrical constraint.

See our reply to your L209-306 content regarding relevant additions to the revised manuscript.

Dacre, H. F., M. K. Hawcroft, M. A. Stringer, and K. I. Hodges, 2012: An Extratropical Cyclone Atlas: A Tool for Illustrating Cyclone Structure and Evolution Characteristics. *Bull. Amer. Meteor. Soc.*, 93, 1497–1502, <https://doi.org/10.1175/BAMS-D-11-00164.1>.

Hawcroft, M. K., L. C. Shaffrey, K. I. Hodges, and H. F. Dacre (2012), How much Northern Hemisphere precipitation is associated with extratropical cyclones?, *Geophys. Res. Lett.*, 39, L24809, doi:[10.1029/2012GL053866](https://doi.org/10.1029/2012GL053866).

Priestley, M. D. K., & Catto, J. L. (2022). Improved representation of extratropical cyclone structure in HighResMIP models. *Geophysical Research Letters*, 49, e2021GL096708. <https://doi.org/10.1029/2021GL096708>

Sinclair, V. A., Rantanen, M., Haapanala, P., Räisänen, J., and Järvinen, H.: The characteristics and structure of extra-tropical cyclones in a warmer climate, *Weather Clim. Dynam.*, 1, 1–25, <https://doi.org/10.5194/wcd-1-1-2020>, 2020.

Reviewer #2 (Remarks to the Author):

Review of manuscript submitted to Communications Earth & Environment

Manuscript identification number: COMMSENV-24-2735-T

Title: Strong polar vortex favoured intense Northern European storminess in February 2022

Authors: Ryan Williams, Amanda Maycock, Vincent Charnay, Jeff Knight, Inna Polichtchouk

Recommendation: Minor Revision

General Comments

The authors analyse the possible links between a stormy period over Western Europe in February 2022 and the strength of the wintertime stratospheric circulation in the previous weeks. While such a link has been previously been hypothesized (e.g. Rupp et al., 2022), the authors quantify based on a multi-model S2S ensemble in how far an intensified stratospheric polar vortex contributed (in a probabilistic way) to a more probable stormy period in February 2022 on the sub-seasonal time scale. I think the authors present a well done and very interesting study which is worth publishing, but several aspects must be better explain before acceptance. The list of comments can be found below. I would be willing to re-review the manuscript upon resubmission if required.

Thank you for taking the time to review our manuscript. We are pleased that you find merit in our work. We provide point-by-point responses here (in blue). Please note that all line (L) numbers quoted correspond to those in the revised track-changed manuscript.

Minor Comments

a) It is not clear while reading the introduction on what are the physical mechanisms leading to serial cyclone clustering and in particular on the crucial importance of having a quasi-stationary large-scale flow for several days for this to happen. See e.g. Dacre and Pinto (2020), page 48, section “dynamics associated with serial cyclone clustering”. I think this information would be very helpful to explain the hypothesis of the paper, how the occurrence of the strong stratospheric vortex may have influenced a steady, stronger than normal westerly flow over the North Atlantic and Western Europe.

Thanks for the suggestion. We agree it would be helpful to explain the relevant dynamical mechanism to place the SPV influence into context. We have added the following to the introduction (L78-80): “The influence of a strong SPV on the jet stream could also affect serial cyclone clustering, which has been linked to a persistent, zonally orientated and intensified jet over the eastern North Atlantic (Pinto et al., 2014).”

b) also introduction, line 34: The definition of a clustering event was first presented in Pinto et al. (2014). Please add the reference here.

Thank you for pointing this out. We have added this earlier reference here.

c) Discussion, lines 170ff: It would be excellent if the authors could elaborate more in how far the February 2022 might be representative from other similar situations or not. Given the

strong importance of the conditions at the tropopause level and internal tropospheric variability of the downward propagation (or not) of signals from the stratosphere, I assume this may be analogue to stratospheric sudden warming and cold waves in Europe: A sudden warming enhances the probability of a cold wave in the following weeks (e.g. Kautz et al., 2020), but it is neither a necessary nor a sufficient condition.

Thank you for your comment. We have added a sentence to the discussion mentioning that our results are specific to 2022 but it would be valuable to conduct the analysis for other similar strong SPV periods. For example, February 2020 had the strongest SPV on record and also exhibited enhanced North Atlantic cyclogenesis. However, there are other observed strong SPV cases which have not coincided with the same tropospheric events. Of course, the observed synoptic dynamics is affected by myriad other factors and drivers than the SPV, and so, similarly to the case of event attribution and climate change, the attribution is probabilistic and not deterministic. It is unclear whether the effects found in this case are representative of other events, so we encourage studies of other periods using our methods.

We have edited the discussion to read (L229-338): “While our results are specific to February 2022, it is plausible that other years with strong SPV conditions could exhibit a similar influence on the North Atlantic storm track. For example, at the time of writing, February 2020 had the strongest monthly SPV since 1979 (Lawrence et al., 2020) and also exhibited intense North Atlantic cyclogenesis (Davies et al., 2021). However, the influence of the SPV on the storm track in February 2022 may have been shaped by other factors related to the tropospheric state at the time, which are captured in the initialised model experiments. It would be insightful to apply the attribution approaches used here to study other periods with strong SPV conditions, to complement composite approaches (Afargan-Gerstman et al., 2024).”

d) Methods, cyclone tracking, line 290ff: It would be important to add some information on how the cyclone statistics produced by this method compared to others (see Neu et al., 2013). Note that Pinto et al. (2016) provided evidence that measures of clustering can be very sensitive to the absolute number of cyclones tracks identified by each tracking method, and thus lead to quite different statistics.

We agree that there may be differences between ETC tracking methods. We chose Ullrich et al. (2021) because it is open source, fully documented in the literature, and because the required input fields were readily available from all the datasets used in the study. Ullrich et al. (2021) show climatological ETC cyclone density in reanalysis data, which compares well with equivalent output from other tracking methods (e.g. compare Fig. 4 of Ullrich et al. (2021) with Fig. 2c of Gramcianinov et al. (2020)).

We have added a couple of sentences in the Methods further expanding on tracking methods and caveating that we use one method in the study:

L423-426: “We note that a variety of methods exist for tracking extratropical cyclones which can produce different results (Neu et al., 2013), including in relation to cyclone clustering (Pinto et al., 2016). However, we use the same method applied to all datasets so the data are directly comparable”.

Gramscianinov et al. (2020) Analysis of Atlantic extratropical storm tracks characteristics in 41 years of ERA5 and CFSR/CFSv2 databases, *Ocean Engineering*, Volume 216, 108111, doi: <https://doi.org/10.1016/j.oceaneng.2020.108111>

e) Methods, SSI, lines 360ff: I cannot follow the justification for the use of a 95. Percentile instead of the commonly used 98. Percentile. In order to justify such a step, the authors would actually need to provide evidence that significant building losses occur already if the 95. Percentile is exceeded in the study and that the loss grows with v^3 above this value. As far as I know, there is no evidence for the above. Please enhance.

Thank you for raising this valid point. We are aware that the 98th percentile is commonly used, and this is because in observation data for Europe, this threshold approximately equals 20 m/s which is the cut off used by insurers for payouts against wind storm damage. The relatively smaller sample size of the ensemble experiments (compared to a longer multi-annual record) meant the 98th percentile could not be defined very accurately and so we opted for a lower, but still extreme, threshold. We note that model biases mean the 98th percentile threshold does not match the insurance standard (20 m/s) in many models and therefore arguably it has less meaning in model experiments. Furthermore, given the cubic relationship of SSI and our lower threshold, the estimates for the relative increase in risk in Table S3 (old Table S4) are actually conservative.

We have added the below couple of sentences to explain our motivation and justification for calculating SSI using v95.

L510-515: “Our choice of v95 was motivated by the desire to look at extreme wind speeds, but also because we are comparing two relatively small samples from the February 2022 simulations and want to ensure the threshold is not determined by one or two outlier points. We note that the choice of a lower percentile threshold, combined with the cubic dependence of SSI on wind speed, would tend to give a conservative estimate of the increased risk of cyclone damage”.

References:

Dacre and Pinto (2020) doi:10.1038/s41612-020-00152-9

Kautz et al. (2020) doi:10.1002/qj.3724

Neu et al. (2013) doi:10.1175/BAMS-D-11-00154.1

Pinto et al. (2014) doi:10.1002/2014JD022305

Pinto et al. (2016) doi:10.3402/tellusa.v68.32204

Rupp et al. (2022) doi:10.1002/2021JD035667

Reviewer #3 (Remarks to the Author):

Comments on “Strong polar vortex favoured intense Northern European storminess in February 2022” by Williams et al.

Summary

Thank you for taking the time to review our manuscript. We provide our point-by-point responses here (in blue). Please note that all line (L) numbers quoted correspond to those in the revised track-changed manuscript.

Using the ERA5 reanalysis and the C3S datasets, and the experiments to explore the possible impact of strong stratospheric polar vortex on the storminess in northern Eurasia. Most of the key findings are shown in the supplementary materials, and the result parts are too simple and short.

The editorial guidelines for manuscripts in Communications Earth and Environment state “As a guide, we recommend that Articles be limited to less than ~5,000 words of main text (the length of the Methods section is not limited and does not count towards the word count).” And necessitate a short letter format. In response to this comment and one from another reviewer, we have brought in some additional figures into the main manuscript (also Table S1 which is new Table 1). Where specific issues of clarity have been raised, we have expanded the text accordingly.

Further, there is still a debate or a concern about the relationship between the stratospheric polar vortex and the storm on the Atlantic-Eurasia sector. Study on the possible impact of the polar vortex on the storm is not consistent between this study and others. It confuses readers whether the strong or the weak polar vortex increases the possibility of the gust and rainfall in Eurasia. Due to those confusions, I suggest a major revision at the present time.

We do not agree that our results are inconsistent with earlier studies on either weak or strong polar vortex influence on European climate. We show a mean positive North Atlantic Oscillation response, the canonical response to a strong SPV and opposite to the canonical response to a weak SPV. This is associated with a **shift** in storminess between northern (NAO+) and southern (NAO-) Europe (e.g., Afargan-Gerstman et al., 2024), so the reviewer needs to consider spatial distribution rather than an overall increase or decrease.

Ours is the first study to our knowledge which explicitly considers cyclone clustering. We must admit that we are not aware of a contentious debate in the literature, in terms of the increased risk of stormy conditions over central and northern Europe following a strong SPV state.

Major comments

1. Most of the key findings are shown in the supplementary materials, and the discussion on the supplementary takes up most of the draft than the existing results concerned Figures 1-4. Key findings should be shown in the main body of the study rather than shown in the supplementary.

Thank you for your comment. You have not specified which results from the supplementary information you feel should be in the main text but we have moved old Fig. S1 into the main text (new Figure 2) added a new figure based on some panels from old Figs. S2, S3, S7 (Figure 3). We think the other figures in the supporting information help to demonstrate our main results are robust to various methodological factors, but are not essential to the overall conclusions and therefore do not need to be in the main text.

Nevertheless, plots which remain in the supplementary information have been reordered in accordance with ordering of necessary mention in the main text and the inclusion of a few additional results into the main manuscript.

2. Some of the previous studies in the introduction link the weak polar vortex and the European storm, while this study links the strong polar vortex and the storm. I am wondering which is true. This study finds that strong polar vortex is more zonally uniform than the weak polar vortex. How does the round strong polar vortex affect the European rainfall and wind gust if not displaced from the Arctic?

Both weak and strong SPV have been linked to changes in the North Atlantic storm track, in particular a northward shift in strong SPV and a southward shift in weak SPV (e.g. Afargan-Gerstman et al., 2024). Strong polar vortex states tend to favour more intense storminess over the UK and Northern Europe, while there is increased storminess over Iberia following weak SPV events.

It is true that the vortex tends to be more zonally symmetric in structure during strong SPV events. Nevertheless, the increased stratospheric zonal winds influence the tropopause region and promotes a strengthening and poleward shift of the tropospheric mid-latitude jet. This in turn favours a poleward and intensified storm track that can impact a large swathe of northern Europe.

3. I can understand the difference between strong and weak polar vortex for C3S models. However, how was the weak polar vortex selected from the model datasets? The weak polar vortex is also misleading due to the fact that the weak polar vortex wind from the C3S models is nearly identical to the climatology wind from ERA5. I do not understand why the weak polar vortex shows the wind identical to the climatology.

As explained in the methods (L317-326), the weak members from the C3S models were chosen as those with extratropical lower stratospheric February zonal winds below the 20th percentile of the ensemble members. As noted in the reply to reviewer 1, it was not expected a priori that the weak members would lie close to the ERA5 climatology and this can rather be viewed as 'chance'. However, this finding indicates the models had skill at a 1–2-month lead time that the SPV would be above average strength (i.e., the weakest members have an average strength and almost none have a below average strength compared to reanalysis).

We have tweaked the text (L321-326) to make clear the match between the weak members and ERA5 was not expected.

4. This study states that the strong polar vortex has a strong impact on the storminess. However, the difference between the strong polar vortex and the climatology or the difference between the OBS experiment and the CLIM experiments is not tested. I wonder the significance level of the strong polar vortex impact on the storm.

We have applied statistical tests to the differences between the experiments for several of the key parameters. This includes bootstrap analysis in old Figure 2d (now Figure 4d), where we estimate shifts in likelihood of a storm of a given intensity between the observed strong SPV state versus climatology. We also apply this in the context of the storm severity index evaluation shown in Table S3 (old Table S4).

The fine details are included in the methods for transparency of how we implemented these statistical tests and we hope that further modifications help to clarify exactly what we did.

5. Another aspect is the physical mechanism between the strong polar vortex and the surface gust. How can we understand the relation between the strong polar vortex and the wind at surface? This study assesses the wind speed difference between the strong and “weak” polar vortex. Consistency of wind direction is not shown.

As mentioned previously, there is an extensive literature on stratosphere-troposphere dynamical coupling demonstrating a connection between the strength of the stratospheric polar vortex and the large-scale North Atlantic circulation (e.g. the NAO, jet stream and storm track) (see e.g., Domeisen and Butler (2020) for a recent review).

We do not understand the comment about wind direction as it does not refer to a specific figure. We compute wind speed gusts as a measure of potentially damaging near-surface winds.

Domeisen, D.I.V., Butler, A.H. Stratospheric drivers of extreme events at the Earth’s surface. *Commun Earth Environ* **1**, 59 (2020). <https://doi.org/10.1038/s43247-020-00060-z>

Other comments

1. L23-25: This is a case study, and the generality of this study is still not known. Since most studies reported the possible link between the weak polar vortex and the extremes in Eurasia, I am still not sure if this case can stand for all strong polar vortex events.

This is an important point, and we are careful to emphasise this is a case study and further work is needed to understand if these findings apply to other strong SPV events. We have added the following to the discussion:

L229-238: While our results are specific to February 2022, it is plausible that other years with strong SPV conditions could exhibit a similar influence on the North Atlantic storm track. For example, at the time of writing, February 2020 had the strongest monthly SPV since 1979 (Lawrence et al., 2020) and also exhibited intense North Atlantic cyclogenesis (Davies et al., 2021). However, the influence of the SPV on the storm track in February 2022 may have been shaped by other factors related to the tropospheric state at the time, which are captured in the initialised model experiments. It would be insightful to apply the

attribution approaches used here to study other periods with strong SPV conditions, to complement composite approaches (Afargan-Gerstman et al., 2024).”

2. L34-35: The 4th is not a reasonable excuse to study this case. I am interested to know if the strong polar vortex is also accompanied with the extremes for the other three cases.

We do not use this as an ‘excuse’ rather that this was a recent impactful event of interest to the research community. We do not claim that a strong SPV would explain all years with high cyclogenesis. Multiple factors could contribute to this which might differ amongst years. However, our specific interest for this year was the coincident occurrence of a strong SPV, similar also to February 2020 when the vortex was strong and there was enhanced cyclogenesis (Davies et al, 2021).

3. L38: What is the state of the Arctic sea ice and the tropical SSTs?

These are initialised in the forecasts and evolve similarly between the two experiments so are unlikely to contribute to any differences we are interpreting to be the influence from the SPV. 2022 was a La Niña winter, which could have been a driver of the strong SPV as noted on L67. Arctic Sea ice extent was 2% below the February average for 1991-2020 <https://climate.copernicus.eu/sea-ice-cover-february-2022>.

4. L67-68: I still do not follow how the strong polar vortex affects the storminess.

See replies to previous comments on the same topic.

5. L84: Positive NAO corresponds to less storms, rather than more storms in midlatitudes.

We are not sure we would agree with this statement. It depends on what is considered mid-latitudes, but there is clear evidence that the poleward shift of the storm track during positive NAO increases storms over northern Europe.

Note that Lockwood et al. (2023) for instance shows that the number of UK winter storms is very strongly correlated (~ 0.85) with the seasonal mean windspeed, which in turn is strongly linked to a positive AO/NAO by construction (i.e., 1st EOF of the mean sea level pressure field). This relationship holds true as the winter mean circulation over the North Atlantic consists of the Azores High and Icelandic Low (prevailing westerly flow).

Lockwood, J. F., Stringer, N., Hodge, K. R., Bett, P. E., Knight, J., Smith, D. et al. (2023). Seasonal prediction of UK mean and extreme winds. *Quarterly Journal of the Royal Meteorological Society*, 149(757), 3477-3489 (2023). <https://doi.org/10.1002/qj.4568>.

6. L121: Not exactly UK, but a wider region.

We appreciate this region is wider than just the UK mainland and also includes surrounding offshore regions (blue box in Figure 1a). The domain does however extent further from the UK upstream (i.e., to the west) as this is the direction from which most cyclones will approach the UK.

7. L130: How does the strong westerly jet promote cyclogenesis?

A stronger jet is associated with higher baroclinicity, e.g. as measured by the Eady growth rate which is proportional to the vertical wind shear, creating conditions that favour the development of baroclinic waves (e.g., Thorncroft et al., 1993).

Furthermore, a persistent, zonally orientated and intensified jet stream over the eastern North Atlantic has been linked to serial cyclone clustering (Pinto et al., 2014).

Thorncroft, C. D., Hoskins, B. J., & McIntyre, M. E.: Two paradigms of baroclinic-wave life-cycle behaviour. *Quarterly Journal of the Royal Meteorological Society*, 119(509), 17-55, <https://doi.org/10.1002/qj.49711950903> (1993).

8. L151: Is there a possibility that the increase in the wind speed is due to the sample bias? I have no idea to what extent this increase is true.

We are unsure what sampling bias you are referring to. We have compared the magnitude of the differences between strong and weak SPV states with noise computed from bootstrap sampling the data.

9. L188: A statistic is possible if you examine the historical records using the ERA5 reanalysis. This is beyond the scope of the current work and will be a topic of future study.

10. Table 1: Where did you show the C3S results? Are the daily data available for the seven seasonal models? We do know which initializations are available and shown.

The C3S results are shown throughout the manuscript (Figs. 3, 4, 5, 6...). The time frequency of the variables and the initialisations used are explained in the section of the manuscript where Table 2 (old Table 1) is referred to.

11. L259: How can we be sure that the data in other members are not problematic?

All the data were carefully checked for missing values. Only these members had a problem which was identified in coordination with the modelling centre who ran the experiments.

12. L265-268: Are the nudging only done between 50-90 hPa? But you define polar vortex at 100 hPa?

Indeed, the full-strength nudging applied in the simulations is only for pressures below 50 hPa (i.e., higher altitudes), with a gradual tapering off in this nudging strength (following a cubic function) from 50 hPa to 90 hPa. For pressures above 90 hPa (i.e., lower altitudes), there is no nudging applied. We refer to the strength of the polar vortex in the lower stratosphere at 100 hPa as it interacts more closely with the troposphere and is more relevant for our case-study investigation. Importantly, this level is not directly impacted by the nudging but is tightly constrained by the nudging applied higher up.

13. L273: This is an ill sentence.

Thanks for spotting this error. It has been fixed.

14. Figure 1: b, what timespan is this histogram is based on? c. monthly maximum is too occasional to explore the impact of strong vortex on wind speed. What about the monthly mean?

The histogram contains each February between 1979 and 2022 ($n = 43$). We have added this to the Figure caption.

Regarding wind gust, we wanted to focus on the maximum wind gust as this is most impact relevant. Nevertheless, we have performed this analysis for the monthly mean wind gust and the signal is qualitatively similar in terms of spatial pattern.

15. Figure 2: Weak and strong polar vortex are too subjective to define here. Methods for d is missing. Please clarify.

The definitions for weak and strong polar vortex are clearly explained in the methods. The explanation and equation for computing the Relative Risk ratio in (d) is now explained under the 'Statistical methods' subsection of the Methods (L517-548).

16. Figure 3: Results are not consistent between C3S models and experiments.

We think the reviewer has not interpreted the legend clearly. The C3S, GloSea6 and IFS members are combined here into two groups: one representing weak SPV and one representing strong SPV. Therefore, there is no problem with consistency.

17. Figure S1: It is weird to see the OBS climatology is equal to weak in a.

Thank you for your observation. It just so happens that the set of weak SPV members taken for the set of forecasts initialised on 1 January are actually of a magnitude comparable to that of ERA5 climatology. The reason for this we can conclude is that the C3S models already had skill at a 1–2-month lead time that the polar vortex was set to become anomalously strong in February 2022.

We have added context of the 10th and 90th percentile values for the month, according to ERA5 climatology (new Fig. 2c). We hope this helps.

18. Figure S2: Monthly or daily data used here?

Thanks for your remark. We calculated monthly averages from all 6-hourly timesteps available for each model. We have now added this detail to the Figure caption.

Reviewer #3 (Remarks to the Author):

Comments on “Strong polar vortex favoured intense Northern European storminess in February 2022” by Williams et al.

We thank reviewer 3 for taking the time to review our paper once more. We provide our point-by-point responses here (in blue). Please note that all line (L) numbers quoted correspond to those in the latest revised track-changed manuscript.

Summary

The largest concerns in my previous review were not seriously considered by the authors. Specifically, the weak polar vortex defined in this study is very different from other studies in literature. The weak polar vortex and the vortex climatology is identical in this study. Therefore, I did not see any scientific value added to existing literature.

We respectfully disagree with this comment and believe the reviewer has misunderstood the focus of our study. Specifically, we focus on the strong polar vortex state in February 2022 and compare this to a counterfactual situation with near climatological conditions. We do not study a weak polar vortex, i.e. where the polar vortex strength is below climatology. We believe the confusion has arisen from our use of the terminology ‘Strong’ and ‘Weak’ to describe the subsets of C3S forecasts. Here, ‘Weak’ refers to the February 2022 C3S forecasts with the weakest polar vortex strength across all ensemble members. As we show, these translate to an approximately climatological polar vortex strength when compared to reanalysis data over many years. Therefore, ‘Weak’ is used in a relative sense applied to the February 2022 forecasts, rather than an absolute sense. We have revised the naming convention of the experiments to avoid confusion using ‘Strong SPV’ and ‘Average SPV’.

Further, the study is focused on a very narrow area, which might not attract wide attention to this study. After careful consideration, I do not recommend publication of this study.

We are unsure what the reviewer means by ‘narrow area’. The case study in our paper constituted a very impactful event that received widespread media and public interest. Ours is the first study to link the strong polar vortex with European cyclone clustering. The implications of the work are directly relevant to the field of sub-seasonal to seasonal forecasting and sectors that rely on this (e.g., transport, insurance). Thus, we are unable to understand the logic that our study does not constitute an important scientific contribution that should be published.

Major comments

1. This study states that both weak and strong polar vortex can cause wide storm activities in Europe. If so, I can not link the storminess to the vortex, because the storm activities do not depend on the polar vortex. If a polar vortex is strong over the Arctic, the weather in Europe is relatively tranquil. However, the authors argue that both strong and weak polar vortex explain the storminess in Europe. I am seriously doubting about those results.

This interpretation of our study is incorrect. We show that in February 2022 there was an enhanced risk of storminess over the UK and Northern Europe under a strong SPV relative to if the polar vortex was under normal conditions. Our study does not present results pertaining to a polar vortex that is below climatological strength. See answer to opening point above. The reviewer's comment "If a polar vortex is strong over the Arctic, the weather in Europe is relatively tranquil" is unsubstantiated and in direct contradiction to the evidence we present in our study, as well as other literature (e.g., Lockwood et al. 2023; Afargan-Gerstman et al. 2024).

2. This is a case study, and a comparison between this case and others is necessary. For example, if the findings in this study is also applied to other cases. The strong polar vortex was also observed in 2020, a review on previous studies is necessary to further confirm finding in this study. I find results in this study might be not consistent with other reported. For example, the strong polar vortex in 2020 was accompanied with an ozone loss over the Arctic and warmer spring. Please refer to several studies here for reference (<https://doi.org/10.1029/2020JD034190>; <https://doi.org/10.1029/2020JD033524>). Although the month focused is a bit different, a review and discussion is required to confirm the consistency or inconsistency between this study and others. The review in this study is too narrow, and a wide review should be performed.

We clearly state within the 'Discussion and conclusions' section of the manuscript that our conclusions apply to the February 2022 case study and that applying our approach to other case studies is warranted. Nevertheless, we argue the findings from this event are worthy of being shared with the community given the interest in extreme events and cyclone clustering.

As noted in our Discussion, the strong polar vortex in February 2020 was also accompanied by a period of intense North Atlantic cyclogenesis and storm clustering. We are unsure why spring-time ozone loss would be relevant to this mid-winter event; the two studies mentioned focussed mainly on S2S predictability of Arctic ozone loss rather than the storm track conditions. Nevertheless, it is interesting to note the positive zonal wind anomaly extending down into the Northern Hemisphere troposphere (~55-80°N) during March 2020 (Fig. 2j in Rao and Garfinkel, 2020), likely reflecting a strengthened mid-latitude jet stream and enhanced propensity for storminess over these latitudes, consistent with our findings for 2022.

3. The structure of this study is also problematic, and key results are not placed in the main body of this study, but in supplementary. The authors argue that the word limited is 5000 words. This limit can include most of figures in the supplementary, and the 5000 word limit should exclude reference lists. Therefore, I am not satisfactory with the revision of this study, although comments from other two reviewers were seriously considered. My comments were not.

Following the previous round of reviews, we carefully considered which material should be included in the main paper. We moved some material into the main paper following the reviewers' suggestions. Now, the Supplementary Information (SI) presents supporting

analyses such as model evaluation and results from the main paper shown for a wider North Atlantic region to test robustness of our findings to the domain choice. We believe our main conclusions are supported by the analysis in the main paper and do not rely on the SI. Including more of the SI in the main paper would impede the flow and likely distract the reader from the main story.

Specific comments

1. The response “We do not agree that our results are inconsistent with earlier studies on either weak or strong polar vortex influence on European climate...” Consistency with Afargan-Gerstman et al., 2024 does not ensure consistency with other literature. Therefore, my comments should be considered again. What aspect is consistent, and what aspect is inconsistent? Please make a careful review.

We are still unsure to which literature reviewer 3 refers to that is inconsistent with our results. We cite a range examples which show that a strong SPV coincides with an NAO+ pattern, a poleward shifted storm track and decrease in cyclone minimum MSLP over the North Atlantic sector. All three aspects are consistent with our results. The exact sentence is included again for reference.

L67-70: “On average, a strong SPV coincides with a positive phase of the North Atlantic Oscillation (NAO)¹³⁻¹⁴ [Ambaum and Hoskins, 2002; Kidston et al., 2015], a poleward shifted storm track¹⁵⁻¹⁶ [Goss et al., 2021; Afargan-Gerstman et al., 2024], and a decrease in average North Atlantic cyclone minimum MSLP¹⁵ [Afargan-Gerstman et al., 2024].”

2. The reply “As explained in the methods (L317-326), the weak members from the C3S models were chosen as those with extratropical lower stratospheric February zonal winds below the 20th percentile of the ensemble members.” I did not agree with this method to choose weak and strong polar vortex. The strong and weak polar vortex should be selected from a complete continuous dataset. For ideal forecasts, the polar vortex should be consistently strong or weak in models. That is, you can not choose weak polar vortex from an ensemble of strong polar vortex. You also can not choose strong polar vortex from an ensemble of weak polar vortex. This method is not reasonable and scientific. To solve this problem, you should use all hindcasts at least 20 or 30 years. The model climatology should be assessed first.

We believe this comment relates to the point raised earlier about a weak polar vortex and the misinterpretation of our study. We do not examine the influence of a weak polar vortex if defined relative to the long-term climatology. Rather, our ‘Weak’ C3S members in the 2022 forecasts compare with near climatological polar vortex strength. Hence, we are isolating the effect of the strong polar vortex in 2022 compared to a counterfactual state with near normal conditions. We have made edits to try and minimise the risk of this confusion including renaming the experiments and no longer referring to ‘Weak’ polar vortex members.

We have revised the relevant sentence in the ‘Introduction’ and added a further sentence (see below).

L82-85: *“This study investigates the role of the strong SPV on the anomalous North Atlantic storm track in February 2022 using operational multi-model ensemble seasonal forecasts (C3S), separated into ensemble members that simulate a strong and average strength SPV (see Methods; Fig. 2a).*

In the ‘Discussion and conclusions’ section, we include the revised sentence below to highlight that there was signal for a stronger than average SPV a month before and that the C3S members corresponding to an average SPV correspond to the weakest 20 % of C3S members:

L205-208: *“While it was not the focus of this study, it is notable that the C3S models initialised on 1 January 2022 were confident the February SPV would be stronger than average, since the average SPV members shown in Figure 2a correspond to the weakest 20 % of members.”*

This is also reiterated in the ‘Methods’, with additional text added in parentheses:

L302-305: *“Whilst this agreement was not expected a priori, it indicates the C3S multi-model forecasts were confident that the SPV would be above average in strength at a 1- to 2-month lead time (highlighted by the fact that the weakest 20 % of members correspond to an average SPV).”*

It is important to note that our approach taken for C3S is not analogous to that employed for composite event studies of strong versus weak polar vortex states (e.g., Afargan-Gerstman et al., 2024), so this needs to be recognised.

3. The reply “We do not understand the comment about wind direction as it does not refer to a specific figure.” If the wind direction is not considered, and the wind direction consistency is not examined, how can you ensure all the response is from the stratospheric polar vortex. This reply is unsatisfactory for reviewer.

The role of the stratospheric polar vortex is isolated by the experiment design, comparing strong polar vortex states with a near-normal polar vortex state. We can therefore be confident the changes in wind gusts result from the difference in polar vortex state, since other factors remain consistent between the two experiments. We do not include information about wind direction because the storm damage index is a function of wind speed not accounting for direction. However, the large-scale circulation changes shown by the MSLP show a positive NAO anomaly which will correspond to enhanced westerly winds in Europe.

4. The reply “L229-238: While our results are specific to February 2022, it is plausible that other years with strong SPV conditions could exhibit a similar influence on the North Atlantic storm track.” Are you sure that your result is consistent with the 2020 case. I do not think so.

The reviewer does not give evidence for why they disagree with our reply, so this is speculative and unsubstantiated. The evidence for a potential link is that February 2020 also exhibited a strong polar vortex and there were three major UK Met Office named storms during February 2020:

Storm Chiara

https://www.metoffice.gov.uk/binaries/content/assets/metofficegovuk/pdf/weather/learn-about/uk-past-events/interesting/2020/2020_02_storm_chiara.pdf

Storm Dennis

https://www.metoffice.gov.uk/binaries/content/assets/metofficegovuk/pdf/weather/learn-about/uk-past-events/interesting/2020/2020_03_storm_dennis.pdf

Storm Jorge

https://www.metoffice.gov.uk/binaries/content/assets/metofficegovuk/pdf/weather/learn-about/uk-past-events/interesting/2020/2020_04_storm_jorge.pdf

Lawrence et al. (2020) provides strong indication for downward coupling from the stratosphere into the troposphere throughout much of February and March 2020, which manifests as a strong NAO+ pattern in terms of the MSLP anomaly field (e.g., Fig. 5a for January-March). However, our use of ‘could exhibit a similar influence’ is very tentative language that does not make a specific claim.

Lawrence, Z. D., Perlwitz, J., Butler, A. H., Manney, G. L., Newman, P. A., Lee, S. H. and Nash, E. R.: The remarkably strong Arctic stratospheric polar vortex of winter 2020: Links to record-breaking Arctic oscillation and ozone loss. *J. Geophys. Res. Atmos.*, **125**(22), e2020JD033271, <https://doi.org/10.1029/2020JD033271> (2020).

5. The reply “We do not claim that a strong SPV would explain all years with high cyclogenesis.” So what is the scientific/practical significance of this study?

The significance of the study is that we show for the first time the unusual cyclonic conditions in February 2022 that caused significant societal impacts can be linked to the strong polar vortex state. As noted above, other years ‘could exhibit a similar influence’ and it would be worthwhile to build on our work to test this assertion.

6. The reply “These are initialised in the forecasts and evolve similarly between the two experiments so are unlikely to contribute to any differences we are interpreting to be the influence from the SPV. 2022 was a La Niña winter, which could have been a driver of the strong SPV as noted on L67. Arctic Sea ice extent was 2% below the February average for 1991-2020 <https://climate.copernicus.eu/sea-ice-cover-february-2022>; I do not agree with this. The model difference might be atmospheric chaos or unstable model error. The oceanic forcing might be important.

We are unsure what the reviewer means by ‘unstable model error’ but we have included results from multiple forecast models to check for robustness across different systems. The use of ensemble forecasts is designed to remove the effects of ‘atmospheric chaos’, or noise, and to extract the signal due the polar vortex state from other sources of variability. The ensemble forecasts are initialised with the same oceanic conditions so this cannot explain the different behaviours of the experiments with a strong polar vortex compared with an average polar vortex.

7. The comment in previous review “5. L84: Positive NAO corresponds to less storms, rather than more storms in midlatitudes.” If this study focuses on the high latitudes or even Arctic, the scientific significance is even weak. Since the Arctic is covered with the polar vortex most of the time, and it should be so (stormy) most of the time. It is not unexpected any more. I did not see any special value of this study.

Our study does not focus on the Arctic.

8. The reply “This is beyond the scope of the current work and will be a topic of future study.” I do not think it is beyond the scope of this study.

The original sentence to which this comment referred to was: *“It is intriguing that both these events occurred in February and future work might examine whether a strong SPV earlier in winter imparts a similar influence on the storm track”.*

Our work concerns the February 2022 case study and therefore other events like the February 2020 case referred to here are beyond the scope of our study.

9. Structure of the paper is disordered. Table 1 and Table 2 should be in order. My comment was not answered directly: “10. Table 1: Where did you show the C3S results? Are the daily data available for the seven seasonal models? We do know which initializations are available and shown.”

So I ask once again: Are the daily data available for the seven seasonal models? We do know which initializations are available and shown.

Table 1 and Table 2 are in order, only Table 1 is shown at the very end (together with the Figures) as is common practice for a paper in peer review. We have since moved Table 2 to the end of the manuscript as well.

The temporal frequency of the variables extracted for the seven seasonal models are specified already in the text:

L278-280: *“Similarly, as for the reanalysis datasets, we retrieved 6-hourly MSLP, 12-hourly 100 hPa u, and daily maximum 10 m wind gust since previous post-processing. Additionally, we acquired monthly 300 hPa u and v and daily total precipitation (mm).”*

For 10 m wind gust and total precipitation, daily data was the highest temporal frequency available. For the other variables, 6-hourly frequency was the highest available from the C3S archive.

All members from the forecasts initialised on 1st January 2022 were used as stated in Methods.

10. The reply “We think the reviewer has not interpreted the legend clearly. The C3S, GloSea6 and IFS members are combined here into two groups: one representing weak SPV and one representing strong SPV. Therefore, there is no problem with consistency.” Where is the updated figure? Where are the weak SPV shown in the revision. Please tell me where you show and what revision did you do? So this comment was overlook again?

We have outlined in the response to earlier comments that we believe there is a misunderstanding that our results do not pertain to the influence of the polar vortex when it is weaker than climatology. We apologise if our response to the previous review contributed to this confusion. 'Weak' is used in a relative sense to describe the lower tail of the C3S polar vortex forecasts. We have changed the names of the experiments to avoid any confusion.